# A functional genomics screen in planarians reveals regulators of whole-brain regeneration

Rachel H Roberts-Galbraith[1]*, John L Brubacher[2], Phillip A Newmark[1]*[†‡]

[1]Department of Cell and Developmental Biology, Howard Hughes Medical Institute, University of Illinois at Urbana-Champaign, Urbana, United States; [2]Department of Biology, Canadian Mennonite University, Winnipeg, Canada

*For correspondence: rhrgalb@
illinois.edu (RHR-G); pnewmark@
morgridge.org (PAN)

Present address: [†]Morgridge
Institute for Research, University
of Wisconsin-Madison, Madison,
United States; [‡]Department of
Zoology, Howard Hughes
Medical Institute, University of
Wisconsin-Madison, Madison,
United States

Competing interests: The
authors declare that no
competing interests exist.

Reviewing editor: Alejandro
Sánchez Alvarado, Stowers
Institute for Medical Research,
United States

**Abstract** Planarians regenerate all body parts after injury, including the central nervous system (CNS). We capitalized on this distinctive trait and completed a gene expression-guided functional screen to identify factors that regulate diverse aspects of neural regeneration in *Schmidtea mediterranea*. Our screen revealed molecules that influence neural cell fates, support the formation of a major connective hub, and promote reestablishment of chemosensory behavior. We also identified genes that encode signaling molecules with roles in head regeneration, including some that are produced in a previously uncharacterized parenchymal population of cells. Finally, we explored genes downregulated during planarian regeneration and characterized, for the first time, glial cells in the planarian CNS that respond to injury by repressing several transcripts. Collectively, our studies revealed diverse molecules and cell types that underlie an animal's ability to regenerate its brain.

## Introduction

An improved understanding of brain regeneration holds the potential to stimulate new therapies geared toward the treatment of stroke, neurological disease, and brain injury. Brain regeneration, as a process, must occur through several interrelated steps: wound signaling and healing; production of new cells by proliferation, dedifferentiation or transdifferentiation; specification of cell lineages of the proper type(s) and ratio(s); organization of new cells in three-dimensional space; differentiation of precursors; reestablishment of connections required for functional restoration; and, finally, cessation of regeneration to avoid overgrowth or damage to nearby healthy tissue. Of these steps, production of new CNS cells has been a major focus. The advent of mammalian stem cell-based approaches and advances in cell reprogramming enabled the creation of new, often functional, neurons in vitro (*Wichterle et al., 2002*; *Zhang et al., 2001*; *Yan et al., 2005*; *Han et al., 2012*; *Vierbuchen et al., 2010*; *Elkabetz and Studer, 2008*; *Piña-Crespo et al., 2012*; *Wernig et al., 2004*). Additionally, the discovery of endogenous stem cell pools in the mammalian brain (*Altman and Das, 1965*; *Eriksson et al., 1998*) raises the possibility of enhancing the brain's ability to create new neurons from within (for a review of this progress, see *Gage and Temple, 2013*). Despite these advances, functional repair in the human CNS remains an extremely ambitious goal and requires the development of new therapies aimed not only at generation of new cells, but also placing and integrating these cells appropriately within a broader regenerative context.

To understand how the cell biological processes of regeneration are regulated and coordinated, we look to organisms that successfully regenerate in nature. While some species, like zebrafish and axolotls have a greater regenerative capacity than do mammals, species with the ability to regenerate an entire brain de novo are rare in the animal kingdom and in the realm of established model

**eLife digest** Animals differ in the extent to which they can regenerate missing body parts after injury. Humans regenerate poorly after many injuries, especially when the brain becomes damaged after stroke, disease or trauma. On the other hand, planarians – small worms that live in fresh water – regenerate exceptionally well. A whole planarian can regenerate from small pieces of tissue.

The ability of planarians to regenerate their nervous system relies on stem cells called neoblasts, which can migrate through the body and divide to replace lost cells. However, the specific mechanisms responsible for regenerating nervous tissue are largely unknown. Roberts-Galbraith et al. carried out a screen to identify genes that tell planarians whether to regenerate a new brain, what cells to make and how to arrange them.

The study revealed over thirty genes that allow planarians to regenerate their brains after their heads have been amputated. These genes play several different roles in the animal. Some of the genes help neoblasts to make decisions about what kinds of cells they should become. One gene is needed to make an important connection in the planarian brain after injury. Another helps to restore the ability of the planarian to sense its food. The experiments also show that some key genes are switched on in a new cell type that might produce signals to support regeneration.

Lastly, Roberts-Galbraith et al. found that the planarian nervous system contains cells called glia. Previous studies have shown that many of the cells in the human brain are glia and that these cells help nerve cells to survive and work properly. The discovery of glia in planarians means that it will be possible to use these worms to study how glia support brain regeneration and how glia themselves are replaced after injury. In the long term, this work might lead to discoveries that shed light on how tissue regeneration could be improved in humans.

organisms (*Bely and Nyberg, 2010*; *Tanaka and Ferretti, 2009*; *Egger et al., 2007*). Of the small number of animals that regenerate their brains after injury, planarians have emerged as a tractable model system in which to study this remarkable process. Scientists have appreciated the extreme capacity for regeneration found in planarians for over two hundred years (*Elliott and Sánchez Alvarado, 2013*). A nascent brain structure appears soon after amputation (*Cebrià et al., 2002b*) and reproducible brain wiring is indicated by full functional recovery, with resumed movement and feeding after one week (*Inoue et al., 2004*). Anatomical studies have revealed details of the structure and molecular complexity of the planarian central nervous system (CNS), which consists of bi-lobed cephalic ganglia (the brain) and two ventral nerve cords (*Figure 1A*; *Agata et al., 1998*). The planarian brain is composed of a diverse array of neural subtypes that produce conserved neurotransmitters and dozens of neuropeptides (*Nishimura et al., 2007a*, *2007b*; *2008a*; *2010*; *2008b*; *Collins et al., 2010*). These cell types are regionally organized, with subtypes of cells appearing medially or in lateral structures called brain branches that project toward the margin of the head (*MacRae, 1967*; *Cebrià et al., 2002a*). Planarians also regenerate missing brain tissue regardless of the amputation plane or the severity of the injury to the CNS (*Reddien and Sánchez Alvarado, 2004*).

In the past two decades, the introduction of molecular techniques and availability of genomic and transcriptomic data (*Newmark et al., 2003*, *Umesono et al., 1997*; *Sánchez Alvarado and Newmark, 1999*; *Robb et al., 2008*) have enabled great progress in understanding the molecular underpinnings of planarian regeneration. The regenerative ability of planarians depends on a pool of stem cells called neoblasts, at least some of which are pluripotent (*Wagner et al., 2011*). This stem cell pool is heterogeneous (*van Wolfswinkel et al., 2014*; *Moritz et al., 2012*; *Hayashi et al., 2010*) and stem cells respond to injury by migrating, proliferating, and producing daughter cells that differentiate to replace all cell types of the body (*Saló and Baguñà, 1985*; *Baguñà et al., 1989*; *Baguñà, 1976*). Additionally, planarians restore their axial polarity after injury; polarity markers reappear early during the regenerative process and are responsible for resetting the body plan (reviewed in *Adell et al., 2010*). Specifically, anterior cues (including Notum and secreted Frizzled-related proteins) inhibit posterior Wnt ligands to enable head and brain regeneration (*Petersen and Reddien, 2011*).

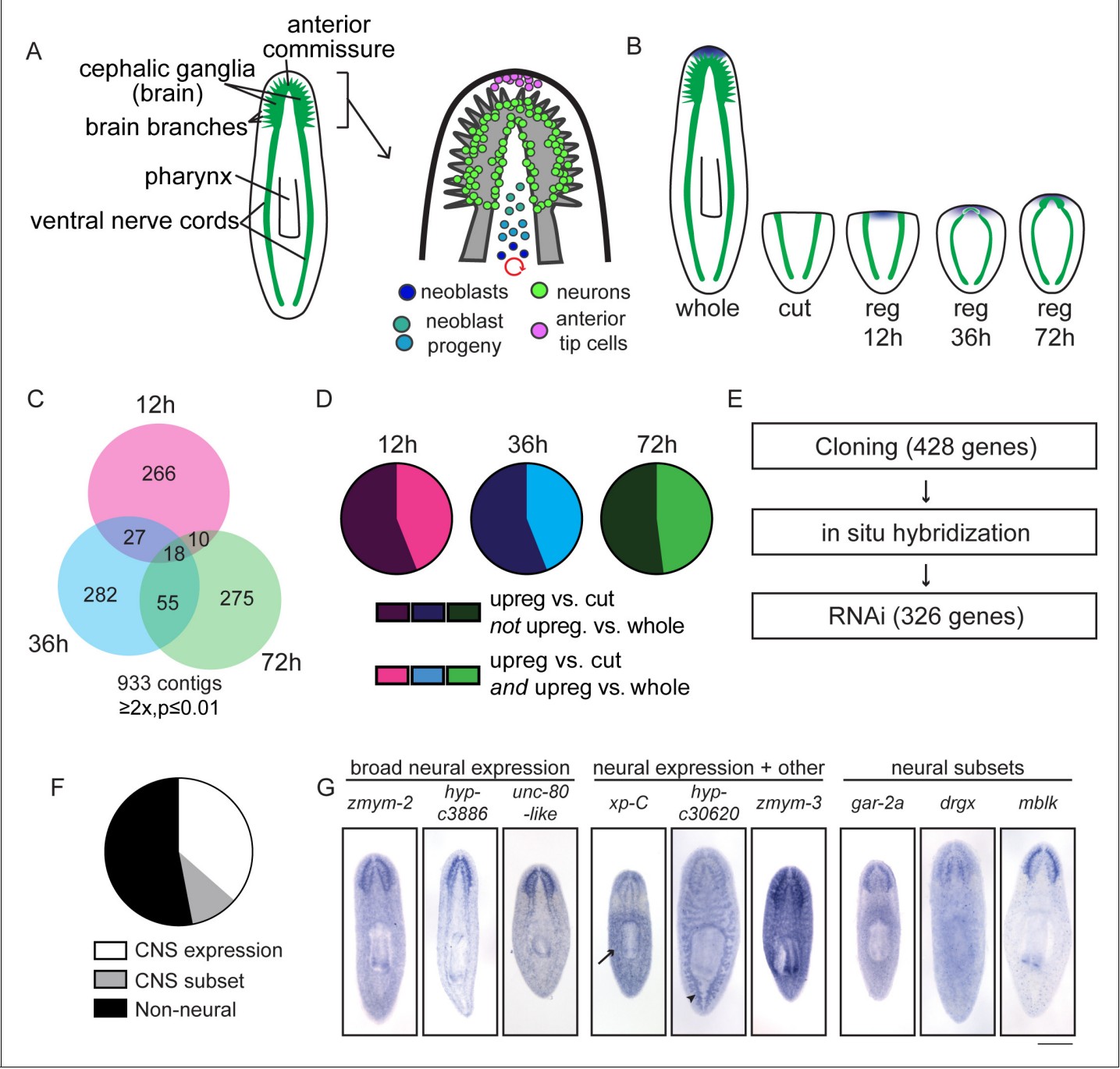

**Figure 1.** A transcriptional view of planarian head regeneration. (**A**) Diagrams depicting the overall organization of the planarian CNS (left) and the principal cell types known to play roles in brain regeneration (right). (**B**) A schematic illustrating the samples used for RNA-Seq analysis. The CNS is shown in green and anterior polarity signals are represented in blue. Whole animals and animals killed immediately after postpharyngeal amputation were used for controls. (**C**) A Venn diagram depicts the 933 contigs upregulated during head regeneration compared to cut controls ($\geq$2x upregulated, p$\leq$0.01). Most identified genes were upregulated to this threshold at only one time point. (**D**) Pie charts showing the proportion of genes upregulated at each time point (vs. cut control) that are also upregulated when compared to uninjured (whole) animals ($\geq$2x upregulated, p$\leq$0.01). The fractions are 44% (12 hr, 36 hr) and 48% (72 hr). (**E**) Workflow depicting our characterization of upregulated genes. (**F**) Pie chart showing the percentage of upregulated genes with neural expression patterns. 362 genes gave clear expression patterns. Of these, 47% had expression in the central nervous system—132 genes with expression in the brain and 38 with expression in a subset of CNS cells. (**G**) Examples of expression patterns (by in situ hybridization, ISH) are shown for genes expressed in the CNS alone, in the CNS and other tissues (arrow indicates the parenchyma for *xp-C* and arrowhead indicates the gut for *hyp-c30620*), and in neural subsets. Scale bar: 500 µm.

*Figure 1 continued on next page*

*Figure 1 continued*

The following figure supplements are available for figure 1:

**Figure supplement 1.** Time courses of anterior regeneration.

**Figure supplement 2.** Expression patterns of genes upregulated during head regeneration.

Planarian brain regeneration requires stem cell activity and proper specification of axial polarity, but beyond these generalities, the mechanisms underlying specific regeneration of the brain remain mysterious. Several regulators of planarian brain regeneration were uncovered through candidate-gene approaches, including axon guidance cues like Slit/Robo and Netrin/DCC as well as transcriptional regulators of certain neural subtypes, including Lhx1/5–1, Coe, and others (*Cebrià and Newmark, 2005, 2007*; *Cebrià et al., 2007*; *Currie and Pearson, 2013*; *Cowles et al., 2013*). Despite the identification of these molecules, we know little about the mechanisms that coordinate cell biological events required for regenerative success. We do not yet understand how neoblasts are directed toward particular neural fates or how cells of different neural subtypes become distributed to ensure proper ratios of diverse cell types. Though polarity cues certainly play a key role, we still lack an understanding of how cells within the organism interpret such cues. Furthermore, the signaling molecules that promote regeneration generally and instruct replacement of specific cell types are largely unknown. While muscle cells produce polarity cues during homeostasis and in response to injury (*Wurtzel et al., 2015*; *Witchley et al., 2013*), it is not yet clear whether they also produce molecules that promote regeneration, or whether other cells carry out or assist in this key signaling role. Finally, once nascent neural tissue is formed, we do not yet understand how it is patterned, reconnected, and restored to function—especially in the context of pre-existing tissue.

To begin answering these important questions, we examined gene expression during the early stages of head regeneration and employed a functional screen to identify key factors in an unbiased manner. We identified several genes required broadly for brain regeneration, including a novel transcriptional regulator of neural progenitors and several putative signaling molecules, some of which are produced by one or more uncharacterized, differentiated cell types in the planarian parenchyma. Furthermore, we identified factors required to promote connectivity and function in specific brain regions. Finally, by investigating genes downregulated during regeneration, we identified a marker that led us to discover a population of cells that we determined to be glia, based on their gene expression and morphology. Taken together, our work uncovered players in diverse aspects of planarian neural regeneration and neurobiology.

## Results

### Identification and characterization of genes upregulated during head regeneration in planarians

To devise an unbiased list of genes that could promote brain regeneration, we first identified transcripts upregulated during planarian head regrowth. We chose an approach in which we amputated planarians postpharyngeally (*Figure 1—figure supplement 1A*), reasoning that this amputation would allow us to identify global gene-expression changes, as well as changes that occur in the regenerating region itself. To identify key time points, we performed several time courses to examine morphological restoration and resumed expression of known brain markers following this particular injury. We examined regenerating animals at 12 hr or 24 hr intervals by immunofluorescence and in situ hybridization (ISH, *Figure 1—figure supplement 1B*). We confirmed reestablishment of anterior polarity by 12 hr, initial neural gene expression at 36 hr, primordial brain formation after 3 days (72 hr), and further maturation (with synapsin staining and broad neural gene expression) at 5 days post-amputation (*Figure 1—figure supplement 1B*). Based on these time courses, we chose 12 hr, 36 hr, and 72 hr as key time points that span the early stages of brain regeneration (*Figure 1B*).

To identify transcripts upregulated during head regeneration, we performed RNA sequencing on regenerating planarians at three time points (12, 36, and 72 hr) as well as whole (non-regenerating)

planarians and planarians immediately after amputation ('cut') as controls (*Supplementary file 1*). Over 900 transcripts were upregulated ($\geq$2x, p$\leq$0.01) at one or more time points in regenerating animals compared to cut (0 hr) controls that were initially identical to the regenerating animals (*Figure 1C* and *Supplementary file 1*). Furthermore, ~45% of transcripts upregulated vs. cut controls at each time point were also upregulated vs. whole animals (*Figure 1D*), indicating that differentially expressed genes do not simply reflect a return to homeostatic gene expression. Our data were also subjected to weighted gene correlation network analysis (*Supplementary file 2*) so that clusters of similarly regulated genes could be identified. Trends for the most abundant clusters of upregulated genes are shown (*Figure 1—figure supplement 2A*); these modules contain known regulators of neural biology and anterior polarity including *coe* and *lhx1/5–1* (module 6), *foxD* (module 7), and *notum* (module 8) (*Petersen and Reddien, 2008*; *Gurley et al., 2008*; *Petersen and Reddien, 2011*; *Koinuma et al., 2003*; *Cowles et al., 2013*; *Currie and Pearson, 2013*).

We next sought to examine upregulated genes in more detail. Of the genes upregulated compared to cut controls, we eliminated transcripts that were very low abundance, parts of repetitive sequences, or housekeeping genes and cloned 428/933 transcripts for further analyses (*Figure 1E*). We examined the gene expression patterns of these genes by ISH and were able to establish expression patterns for ~85% of them (362/428). We hypothesized that our dataset would be enriched in genes expressed in the CNS and, indeed, found that 47% of genes with clear expression patterns showed enrichment in the CNS (170/362, *Figure 1F–G*). Of the 170 genes with CNS expression, 132 were expressed broadly and 38 showed enrichment in subsets of CNS cells (*Figure 1F–G*). Additionally, genes expressed in the CNS were often expressed elsewhere, for example in the parenchyma or in the intestine (*Figure 1G*). Of upregulated genes with detectable expression patterns, we also found that 9% showed enriched expression in the head (*Figure 1—figure supplement 2B–C*) and 17% were expressed in the parenchyma, some in a pattern similar to neoblast genes (*Figure 1—figure supplement 2D–E*). Additional genes were expressed in tissue-specific patterns that included the pharynx, intestine, protonephridia, epithelium, and eyespots (*Figure 1—figure supplement 2F–G*). Some non-CNS expression patterns could still reflect neural tissue in the pharynx, body wall, or eyes, but we have not investigated neural regeneration outside the CNS at this point. However, the variety of expression patterns reflects the diverse physiological changes that occur concurrently during head regeneration (*Supplementary file 3A*).

## An unbiased functional screen reveals genes with roles in planarian brain regeneration

To determine whether the upregulated genes promote brain regeneration, we performed RNA interference (RNAi) experiments to knock down 326 of the upregulated transcripts (*Figure 2A*). These genes included all those enriched in the CNS, head, or parenchyma, as well as a subset of genes with other expression patterns or for which no pattern was detected. After RNAi we examined brain regeneration by performing ISH to detect *choline acetyltransferase (ChAT)* mRNA, which is expressed in an abundant cell type in the CNS (*Nishimura et al., 2010*). We validated genes that caused a small brain upon regeneration by repeating RNAi and ISH, this time quantifying brain area after regeneration (*Figure 2B*, *Supplementary file 3B*). Of the genes screened, 9.2% (30/326) caused a significant reduction in brain area after RNAi (*Figure 2C*). A wide range of severity was observed for genes associated with this phenotype, with relative brain areas ranging from 16–80% of controls. RNAi targeting two additional genes caused nervous system regeneration phenotypes without a small brain (*Supplementary file 3B*): *sp6-9(RNAi)* caused defects in eye regeneration (*Lapan and Reddien, 2011*) and *arrowhead(RNAi)* caused defects at the midline of the brain which will be described below. If RNAi animals showed gross phenotypes like lysis or curling prior to amputation or regeneration, they were killed and fixed when a phenotype was first observed (*Supplementary file 3C*, *Figure 2—figure supplement 2*).

Genes associated with CNS-regeneration phenotypes were expressed in a variety of patterns, including neural, parenchymal, and ubiquitous expression (*Figures 3–4*, *Figure 2—figure supplement 1A*). Most were upregulated in the anterior-most tissue in regenerating tail fragments (*Figures 3–4*, *Figure 2—figure supplement 1A*), though patterns ranged from small subsets of cells (e.g., *AADC*, *Figure 2—figure supplement 1A*) to broad upregulation throughout the remaining tissue (e.g., *tRNA synthase*, *Figure 2—figure supplement 1A*). A prior study examined the expression of genes in stem cells and their progeny (*Labbé et al., 2012*), and a number of genes identified in our

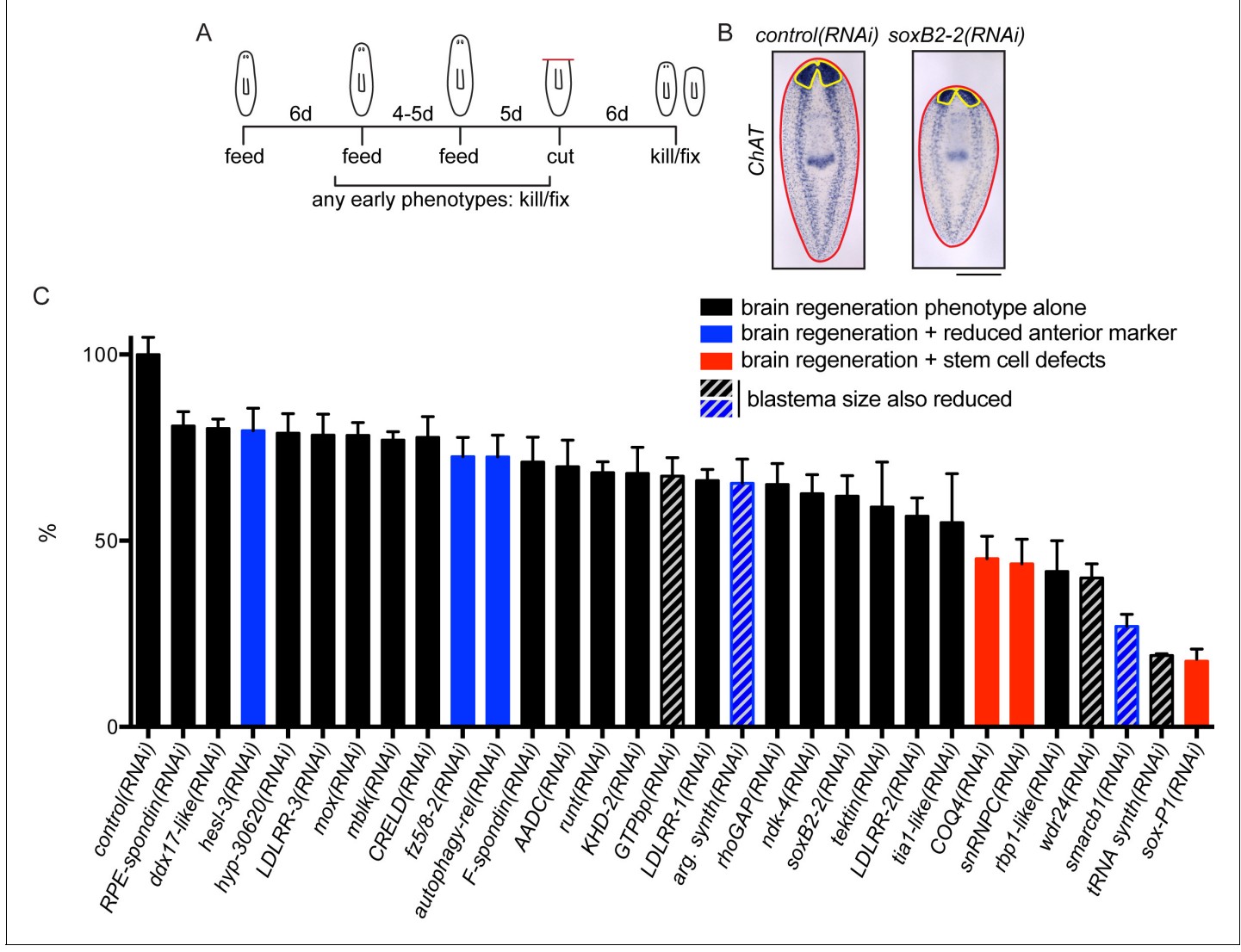

**Figure 2.** A screen for genes required for regeneration of the planarian brain. (**A**) A diagram depicting the RNAi protocol used for our functional screen. For each of the 326 genes analyzed in our study, dsRNA was synthesized and fed to animals three times over 10–11 days. Five days after the final dsRNA feeding, animals were cut prepharyngeally and allowed to regenerate for 6 days. The animals were then killed and fixed for ISH to visualize the CNS. Animals were also monitored for behavioral defects prior to the termination of the experiment. When animals manifested phenotypes earlier than the conclusion of the experiment (e.g., lysis or curling), they were fixed when such a phenotype was first observed. (**B**) Example of RNAi animals used for brain area quantification. *Control(RNAi)* and *soxB2-2(RNAi)* animals were subjected to ISH with the *choline acetyltransferase (ChAT)* probe. After imaging, the body area (red line) and brain area (yellow line) were calculated for each sample. (**C**) Brain area/body area ratios were averaged across ~8 worms per sample and were normalized to *control(RNAi)* animals. All 30 genes for which RNAi caused a significant reduction in brain area are shown here, with error bars representing SEM. Genes for which RNAi also caused reduction in an anterior marker are indicated with blue bars. Red bars indicate genes for which RNAi also affected stem cell maintenance. Bars with grey diagonal lines indicate genes for which RNAi also caused small blastemas after amputation. Scale bar: 500 µm.

The following figure supplements are available for figure 2:

**Figure supplement 1.** Further analysis of genes with functions in CNS regeneration.

**Figure supplement 2.** Upregulated genes play roles in tissue maintenance and stem cell biology.

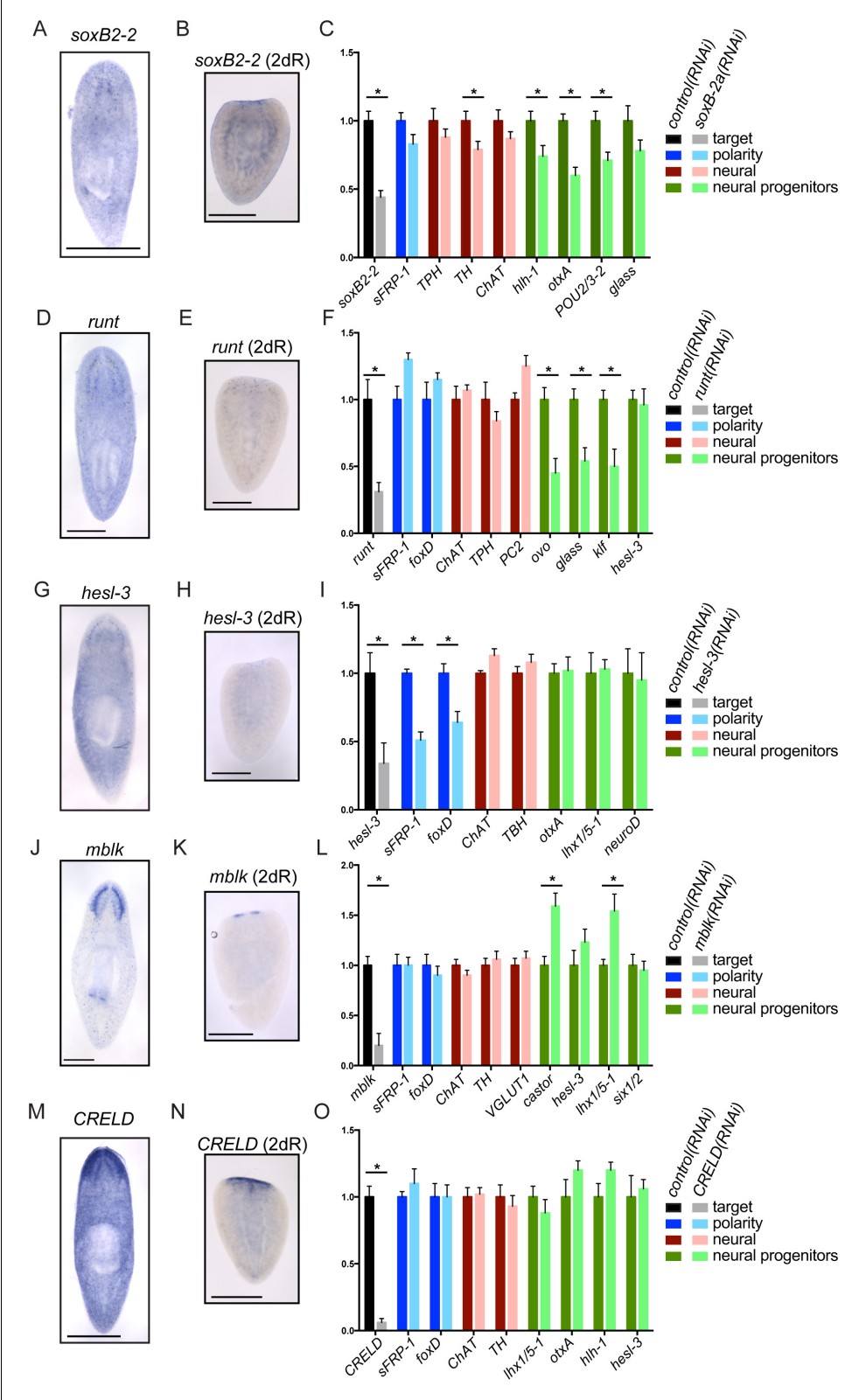

**Figure 3.** Genes important for brain regeneration exert their effects through multiple mechanisms. (**A**) Expression of *soxB2-2* by ISH during homeostasis. *soxB2-2* cells are scarce, but mostly localized near the brain. (**B**) *soxB2-2* expression was also determined by ISH in animals cut postpharyngeally and allowed to regenerate for two days. (**C**) RNA was prepared from whole *control(RNAi)* and *soxB2-2(RNAi)* animals at the end of a treatment course depicted in **Figure 2A**. qPCR was used to examine the expression of 134 genes after RNAi. Selected qPCR results are shown here.

*Figure 3 continued*

The bar graph is color coded to indicate the expression of the gene targeted by RNAi (black/grey), anterior polarity genes (blue), genes important for neurotransmitter biogenesis (red/pink), and genes with suspected roles in neural differentiation or neural progenitors (dark/light green). *soxB2-2* was knocked down to 44% of *control(RNAi)* levels after RNAi. Significant decreases in *hlh-1*, *otxA*, and *POU2/3-2* were also seen after *soxB2-2(RNAi)*. Similarly, expression patterns and downstream effects were determined for *runt* (D–F), *hesl-3* (G–I), *mblk* (J–L), and *CRELD* (M–O). *runt* also affected neural progenitor genes, while *hesl-3* primarily affected polarity genes. No downregulated genes were identified after *mblk(RNAi)* or *CRELD(RNAi)*. Targets were knocked down to 31% (*runt*), 34% (*hesl-3*), 20% (*mblk*) and 6% (*CRELD*) of *control(RNAi)* levels. Asterisks mark genes with a significant difference in expression (p≤0.05) after RNAi. Scale bars: 500 μm.

The following figure supplement is available for figure 3:

**Figure supplement 1.** Upregulated genes play roles in diverse processes to promote regeneration.

functional screen are also expressed in these cell populations (*Supplementary file 3B*). Genes with roles in regeneration belonged to several expression modules in our clustering analysis (*Supplementary file 2* and *Supplementary file 3B*) with module 7 appearing most frequently (7/30, *Figure 1—figure supplement 2A*). This module includes 600 genes that are modestly upregulated at 12 hr and more dramatically so at 36 and 72 hr post-amputation (*Supplementary file 2*). The spatiotemporal diversity of gene expression patterns suggests that these genes likely affect regeneration through multiple mechanisms.

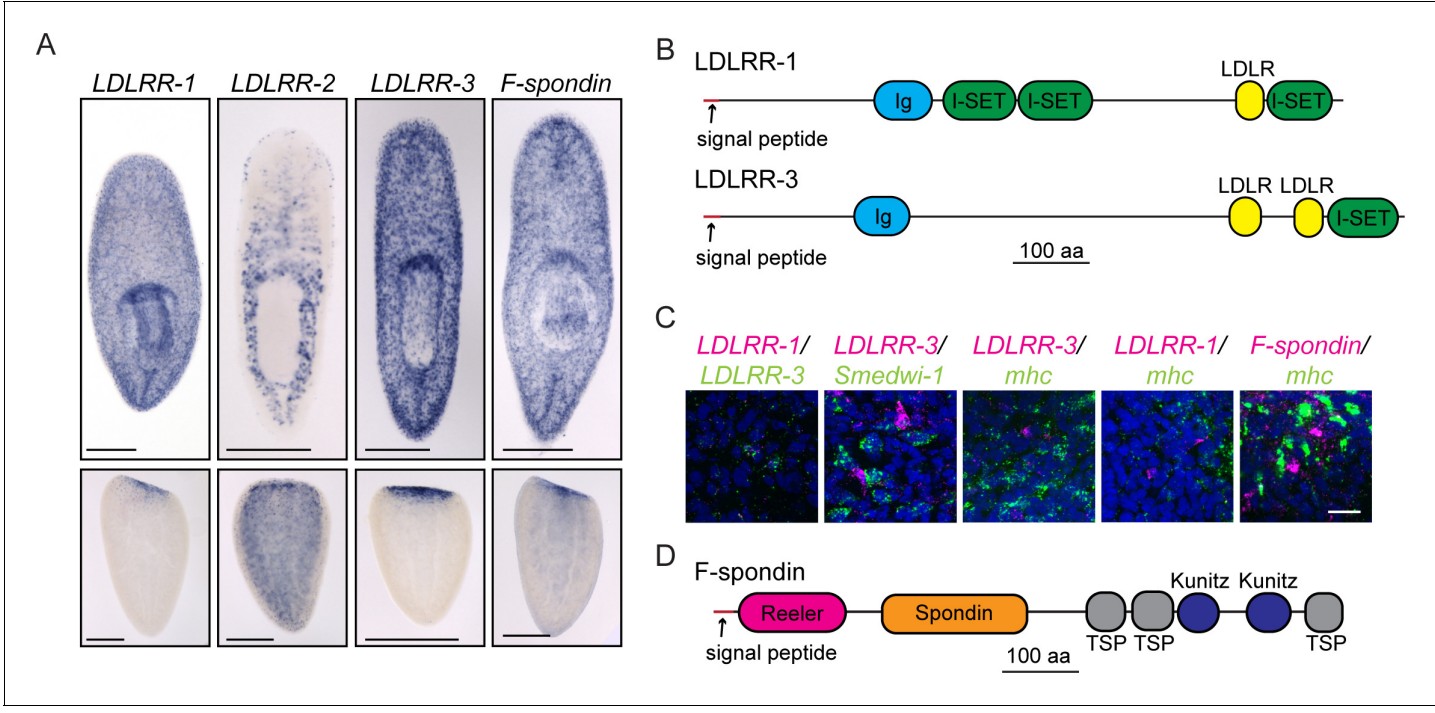

**Figure 4.** Secreted molecules with roles in planarian brain regeneration. (A) Expression patterns of *LDLRR-1*, *LDLRR-2*, *LDLRR-3*, and *F-spondin* were determined by ISH for uninjured worms (top) and tail fragments two days after amputation (bottom). (B) The domain architectures of LDLRR-1 and LDLRR-3. (C) Coexpression of the indicated genes by fluorescent in situ hybridization (FISH). All *LDLRR-3*[+] cells express *LDLRR-1* (16/16), but only about half of *LDLRR-1*[+] cells express *LDLRR-3* (16/31). *LDLRR-3*[+] cells do not express *smedwi-1* (0/34), although these cells are often found in proximity to one another. *LDLRR-1*, *LDLRR-3*, and *F-spondin* are not expressed in *mhc*[+] muscle cells (0/8, 0/23, and 0/6 respectively). *LDLRR* FISH experiments were performed on whole worms and *F-spondin* FISH was performed on 2d regenerating animals. (D) The domain architecture of F-spondin. Scale bars: 500 μm (A), 20 μm (C).

The following figure supplement is available for figure 4:

**Figure supplement 1.** Parenchymal genes are irradiation-insensitive and do not influence wound response.

As noted above, successful brain regeneration depends on several interrelated processes, not all of which are specific to the CNS. Thus, to distinguish between target genes whose knockdown specifically affected CNS regeneration and those which had indirect effects through other mechanisms, we repeated RNAi experiments and examined effects of each of the 30 genes on anterior polarity, stem cell maintenance, and regeneration of tail tissue. First, we examined the anterior marker *sFRP-1* (*Petersen and Reddien, 2008*; *Gurley et al., 2008*) after RNAi of each gene. RNAi knockdown of 5 genes (*argininosuccinate synthase*, *autophagy-related 13*, *smarcb1*, *fzd-5/8–2*, and *hesl-3*) reduced *sFRP-1* expression during regeneration, with reductions ranging from minor to severe (*Figure 2C*-blue bars, *Figure 2—figure supplement 1B*). Thus, for these five genes, the small brain RNAi phenotype could result from insufficient reestablishment of anterior identity. We also investigated whether the stem cell pool was affected after RNAi knockdown of these 30 genes using the neoblast marker *smedwi-1* (*Reddien et al., 2005*). We determined that knockdown of three genes caused a reduction in *smedwi-1* signal: *snRNPC* and *rbp-1-like*, which have not been described, and the previously identified gene *soxP-1* (*Figure 2C*-red bars, *Figure 2—figure supplement 1C*; *Wagner et al., 2012*; *Wenemoser et al., 2012*). We concluded that RNAi of these three genes led to brain regeneration phenotypes secondary to stem cell defects. Finally, to confirm that brain regeneration phenotypes did not merely result from a general impairment in blastema formation (i.e. general defects in neoblast proliferation or migration), we repeated each RNAi experiment and amputated animals anterior and posterior to the pharynx and subsequently measured blastema size after six days. RNAi of five genes (*tRNA synthase*, *GTPbp*, *argininosuccinate synthase*, *wdr24*, and *smarcb1*) caused smaller blastemas after regeneration and only *GTPbp(RNAi)* had an anterior-specific reduced blastema size (*Supplementary file 3B*). Thus, these five genes likely play a role in regeneration that is not specific to the CNS (*Figure 2C*-gray diagonal lines). RNAi of the remaining 19 genes caused no overt polarity or stem cell defects by ISH and no evidence of generally reduced regenerative activity (*Figure 2C*-black bars). We thus prioritized these genes for further study, as these represent the most promising candidates for factors specifically influencing brain regeneration.

Though our current work is focused on regeneration of the planarian CNS, the functional screen we performed resulted in the identification of 26 additional genes for which RNAi caused defects during homeostasis and/or regeneration (*Supplementary file 3C*, *Figure 2—figure supplement 2*). RNAi targeting 12 genes caused head regression and/or curling, while knockdown of 7 genes caused general lysis (*Supplementary file 3C*, *Figure 2—figure supplement 2A*). We used ISH with the *smedwi-1* probe and detected loss or reduction of neoblasts after knockdown of 13 of these genes (*Supplementary file 3C*, *Figure 2—figure supplement 2B–C*); in many of these cases (e.g., *zfp-4*), the causative gene had not been implicated in planarian neoblast biology. As expected, most genes identified in this portion of the screen showed ubiquitous or parenchymal expression patterns (*Figure 2—figure supplement 2D*) with enrichment in stem cells and/or stem cell progeny (*Labbé et al., 2012*; *Supplementary file 3C*). Other knockdowns (e.g., *exo70*, *hnRNPK*, and *smarcc2*) caused tissue homeostasis phenotypes despite the presence of stem cells (*Figure 2—*

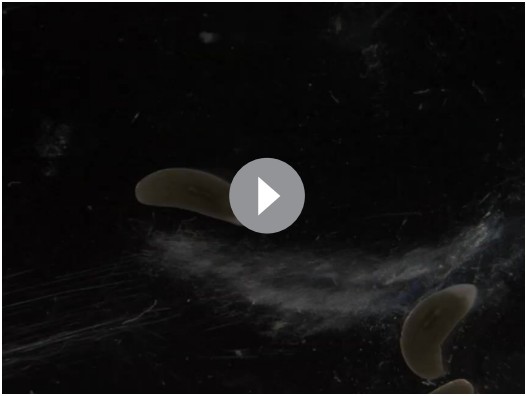

**Video 1.** Live imaging of *control(RNAi)* animals.

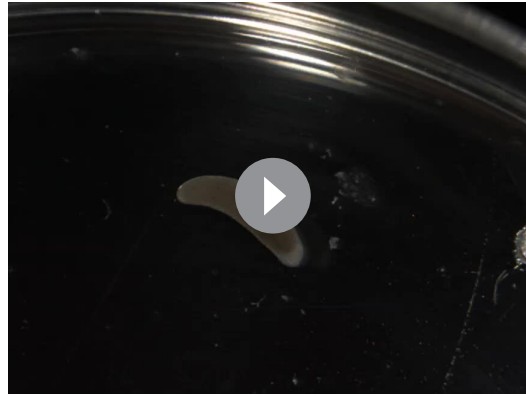

**Video 2.** Live imaging of *soxB2-2(RNAi)* animals.

*figure supplement 2C*), suggesting potential roles for these genes in proper stem cell function, tissue maintenance, or differentiation. One of these genes (*hnRNPK*) is required for pharynx regeneration (*Adler et al., 2014*) and here we show that it is important for head maintenance and regeneration. Therefore, *hnRNPK* could have a broad role in the maintenance or renewal of differentiated cell types. RNAi of one gene, *jouberin*, also caused bloating reminiscent of protonephridial defects (*Figure 2—figure supplement 2A*). All other genes had no detectable phenotypes in our assays (*Supplementary file 3D*).

## Upregulated genes promote brain regeneration through diverse mechanisms

We investigated genes important for brain regeneration to determine whether we could assign them to known pathways or processes. To determine which downstream cells or pathways are affected, we repeated RNAi experiments for five genes (*soxB2-2, runt, hesl-3, mblk,* and *CRELD*) and used quantitative RT-PCR (qPCR) to examine the expression of ~134 genes after the 6-day regeneration period (*Supplementary file 3E–F*). The genes chosen for qPCR analysis represent neural markers (including neuropeptide genes and genes important for neurotransmitter production), markers of neural progenitors (genes expressed in differentiating neoblasts and also associated with neural expression or differentiation), and polarity markers. We also included neoblast transcripts and genes expressed in progenitors of other tissues (protonephridia, intestine, etc.) to rule out general roles in stem cell maintenance or differentiation (*Supplementary file 3E–F*).

### Specification of neuronal progenitors

Importantly, several genes that are expressed in a small number of cells still caused significant reduction of brain size after RNAi. We reasoned that some of these genes might be influencing a subset of neural cell types or progenitors. We began with *soxB2-2*, which encodes a SOX family transcription factor. *soxB2-2* was expressed in cells of the head in whole animals (*Figure 3A*) and is expressed in a small number of cells in the anterior of regenerating planarians (*Figure 3B*). A prior transcriptomic study found that *soxB2-2* was expressed in stem cells and stem cell progeny (*Labbé et al., 2012*). Importantly, *soxB2-2(RNAi)* animals showed a small brain phenotype (*Figure 2B–C*) and also a reduced response to vibrations in their environment (*Videos 1–2*). After *soxB2-2(RNAi)*, we did not detect any defects in anterior polarity by ISH (data not shown) or qPCR (*Figure 3C*). The only neural marker to be significantly downregulated after *soxB2-2(RNAi)* was *tyrosine hydroxylase (TH)* (*Figure 3C*), though loss of neural subsets (rather than whole classes) might be difficult to detect with this methodology. However, several neural progenitor genes were consistently downregulated after *soxB2-2(RNAi)*, including significant downregulation of *hlh-1, otxA,* and *POU2/3–2* (*Figure 3C*). Each of these genes is expressed in a subset of cells in the planarian brain and in neoblast progeny (*Figure 3—figure supplement 1A*; *Labbé et al., 2012*). We confirmed that *hlh-1* signal is reduced by ISH after *soxB2-2(RNAi)* (*Figure 3—figure supplement 1B*). We therefore conclude that SoxB2-2 plays a role in promoting expression of a neural progenitor gene during the course of brain regeneration.

Another gene required during brain regeneration is *runt*, which encodes a transcription factor expressed during homeostasis in the brain and during regeneration throughout the blastema (*Figure 3D–E*) in neoblasts and their progeny (*Labbé et al., 2012*). *runt* was previously identified as a wound-induced gene required for specification of *sp6-9*[+] and *AP2*[+] progenitors and for proper regeneration of cells of the planarian eye (*Wenemoser et al., 2012*; *Sandmann et al., 2011*). We also found that knockdown of *runt* leads to reduced levels of the neural progenitor markers *ovo, glass,* and *klf* (*Figure 3F*), without influencing anterior fate or the broad expression of neural genes. Taken together, these results demonstrate that *runt* is important for directing expression of several neural progenitor genes that promote cell fates in the eye (*sp6-9*[+] and *ovo*) and beyond. *soxB2-2* and *runt* affect distinct progenitor genes, making it likely that the two transcription factors influence different fate choices.

### Influence on anterior polarity and neural patterning

Like *soxB2-2* and *runt*, the upregulated *hes/hey*-related gene, *hesl-3,* is also expressed in presumptive neural progenitors and cells of the brain (*Cowles et al., 2013*; *Figure 3G–H*). However, in contrast to *soxB2-2* and *runt, hesl-3(RNAi)* results in cyclopia and a small brain fused at the midline

(*Cowles et al., 2013*; *Figure 2C*, *Figure 3—figure supplement 1C*). When we examined gene expression after *hesl-3(RNAi)*, the primary effect we observed was downregulation of anterior polarity markers *sFRP-1* and *foxD* (*Figure 3I*, *Figure 2—figure supplement 1B*, *Figure 3—figure supplement 1D*). We were unable to detect decreased expression of neural markers or transcripts marking neural progenitors. Thus, we hypothesize that *hesl-3* acts through a distinct mechanism during head regeneration.

### Novel mechanisms

Finally, two genes caused small brain regeneration phenotypes without an obvious loss of anterior polarity, neural, or neural progenitor markers. One of these genes, *mblk*, is expressed in the brain branches of the planarian CNS (*Figure 3J–K*) and is related to the *mushroom body large-type Kenyon cell-specific (mblk)* gene, which encodes a putative transcription factor in honeybees (*Takeuchi et al., 2001*). *mblk* is not expressed in neoblasts, but in neoblast progeny and differentiated cells (*Labbé et al., 2012*). After *mblk(RNAi)*, we observed no changes in anterior genes or neural markers, and progenitor genes were either unaffected or even upregulated (*Figure 3L*). Thus, *mblk(RNAi)* causes a small brain phenotype via an unknown mechanism that might involve an imbalance of cell types or failure in terminal differentiation. The second gene, *CRELD (cysteine-rich egf-like domain)*, encodes a putative secreted or transmembrane protein (*Figure 3—figure supplement 1E*) and is expressed in the planarian brain and head, with anterior upregulation in regenerating worms (*Figure 3M–N*). We were unable to detect changes in polarity markers, neural genes, or progenitor genes after *CRELD(RNAi)* (*Figure 3O*), despite a small brain phenotype after this treatment (*Figure 2C*). Both *mblk* and *CRELD* are conserved among metazoa, but their homologs in other animals have not been well characterized. Thus, our screen identified conserved genes that regulate brain regeneration through mechanisms that we cannot yet decipher using available markers.

## A differentiated cell type in the parenchyma promotes planarian regeneration

One of the biggest outstanding questions in understanding planarian regeneration is: what molecular cues regulate the process? Among the 30 genes identified in our functional analysis as being required for proper brain regeneration, several genes (including *CRELD*) encoded putative signaling molecules. Of these genes, a few are expressed in differentiated cells (*Supplementary file 3B*; *Labbé et al., 2012*). One such group of genes is the *LDLRR (low density lipoprotein receptor-related)* family. *LDLRR-1* and *LDLRR-2* were previously identified in an analysis of wound-response genes (*Wenemoser et al., 2012*). *LDLRR-1* is expressed in the parenchyma, while *LDLRR-2* is expressed in a subset of intestinal cells and in epithelial cells, particularly in the head (*Figure 4A*). Both show redistribution of expression after injury, with punctate expression in the blastema during regeneration (*Figure 4A*). We also identified a gene with similarity to *LDLRR-1* in our functional screen, which we have named *LDLRR-3*; this gene is also expressed in a pattern resembling that of *LDLRR-1* (*Figure 4A*). *LDLRR-1* was so named for an LDLR (Low Density Lipoprotein Receptor) domain (*Wenemoser et al., 2012*); however, the current gene model for LDLRR-1 includes a signal peptide sequence but no transmembrane domain, suggesting that LDLRR-1 is likely to be a secreted molecule or to remain attached to the cell surface by other means (*Figure 4B*). We were unable to detect known domains within the existing LDLRR-2 sequence, but LDLRR-3 has a domain architecture similar to LDLRR-1 and also lacks a transmembrane domain (*Figure 4B*). Furthermore, *LDLRR-1* and *LDLRR-3* share closest similarity with proteins of the Heparan Sulfate Proteoglycan (HSPG) family (*Supplementary file 3G*). HSPGs play diverse roles in regulation of the extracellular matrix and signaling (for review, see *Lin, 2004*). Thus, the planarian family of LDLRR molecules could serve to modulate signaling in a way that promotes regeneration.

We asked whether *LDLRR-3* is expressed in cell types already known to play key roles during regeneration. *LDLRR-3* is not irradiation sensitive (*Figure 4—figure supplement 1A*; *Labbé et al., 2012*) and we confirmed that *LDLRR-3*$^+$ cells in the parenchyma are distinct from (but sometimes adjacent to) *smedwi-1*$^+$ stem cells (*Figure 4C*). Because muscle cells secrete position-control factors that set up the body axis of planarians (*Witchley et al., 2013*), we tested whether LDLRR-3 could be produced by muscle cells. We found that *LDLRR-3*$^+$ cells do not express the muscle marker *mhc* (*myosin heavy chain*; *Figure 4C*; *Kobayashi et al., 1998*); thus, *LDLRR-3* is expressed in

differentiated, non-muscle cells to promote regeneration. Furthermore, while *LDLRR-1* and *LDLRR-3* are expressed in similar patterns (*Figure 4A*), their expression is only partly overlapping (*Figure 4C*); all *LDLRR-3*$^+$ cells express *LDLRR-1*, but only half of *LDLRR-1*$^+$ cells express *LDLRR-3*. As expected based on this finding, *LDLRR-1* is also expressed in differentiated, *mhc*$^-$ cells (*Figure 4C*; *Labbé et al., 2012*). Because *LDLRR-1* was identified as a wound response gene (*Wenemoser et al., 2012*), we examined other wound response genes (*follistatin, jun, inhibin,* and *runt*) in *LDLRR-1* (*RNAi*) and *LDLRR-3(RNAi)* animals. We were not able to observe any diminished wound response after these perturbations (*Figure 4—figure supplement 1B–C*).

In addition to the *LDLRR* family, a gene encoding planarian *F-spondin* was identified as important for CNS regeneration in our functional screen. *F-spondin* is also expressed in parenchymal, differentiated cells that are not *mhc*$^+$ (*Figure 4A,C, Figure 4—figure supplement 1A*). Furthermore, the predicted planarian F-spondin protein possesses a signal peptide but lacks a transmembrane domain (*Figure 4D*), suggesting that it is secreted, like its homologs in other species (*Feinstein and Klar, 2004*). We thus conclude that one or more parenchymal cell types produce cues (including planarian LDLRR and F-spondin proteins) that promote regeneration of the CNS and/or regeneration more generally.

## Arrowhead promotes reconnection of the brain lobes during regeneration

In addition to exploring genes that are required for brain regeneration, we were interested in examining genes that function in more specific aspects of neural architecture. We hypothesized that regionally expressed genes might affect local connectivity or function in the regenerating planarian CNS. One subset of genes upregulated during regeneration was expressed medially in the planarian brain (*Figure 5A*). RNAi targeting these genes did not cause small brain phenotypes (data not shown). However, we did detect a specific phenotype after targeting the planarian homolog of *arrowhead*, a gene that encodes a LIM-homeodomain transcription factor (*Curtiss and Heilig, 1995*, *1997*). Planarian *arrowhead* is expressed in neurons, at least some of which are *ChAT*$^+$ (*Figure 5A–B*). Despite normal brain size, *arrowhead(RNAi)* animals regenerated with a gap between the lobes of the cephalic ganglia (*Figure 5C*, arrows). The medial gap was more clearly visualized with an anti-synapsin antibody that stains the planarian neuropil, a region rich in axons, dendrites, and synapses (*Figure 5D–E*). The anterior commissure is the largest commissure in the planarian CNS and connects the two lobes of the cephalic ganglia (*Figure 1A*). Many axons cross the midline at the anterior commissure, including photoreceptor axons (*Agata et al., 1998*). We detected photoreceptor neurons using an anti-arrestin antibody (VC-1; *Agata et al., 1998*) and determined that photoreceptor axons appeared frayed and/or failed to cross the midline in a majority of *arrowhead(RNAi)* animals (*Figure 5F–G*). Failure of photoreceptor axons to cross at the anterior commissure was never detected after *control (RNAi)* (*Figure 5F–G*). One factor previously shown to regulate axon guidance in planarians is *slit* (*Cebrià et al., 2007*). While some *arrowhead*$^+$ cells are *slit*$^+$, the overall *slit* expression pattern was unaffected after *arrowhead(RNAi)* (*Figure 5—figure supplement 1A,B*). Taken together, our results suggest that *arrowhead* regulates cells or factors that drive reconnection of the planarian brain lobes and organization of newly regenerated axons at the anterior commissure.

## Planarian brain branches play a role in chemosensation

As described previously, brain regeneration involves not only the production of properly patterned tissue, but also a restoration of function. Planarian brain branches have been ascribed functions in mechanosensing and/or chemosensing (*Oviedo et al., 2003*, *Umesono et al., 1997*, *Nakazawa et al., 2003*) and the auricles at the edge of the head of other planarian species have been shown to be important for chemosensory behavior (*Koehler, 1932*, *Farnesi and Tei, 1980*). The expression of *CNG1*, which encodes a cyclic nucleotide-gated (CNG) channel, as well as other homologs of olfactory signaling proteins in the brain branches (*Figure 6A*), supported the hypothesis that these structures mediate functions specific to chemosensing. We observed a particularly interesting expression pattern for *CNG1*, which was expressed in regions at the very edge of the head (*Figure 6A–B*), a location expected for cells serving in a sensory capacity. CNG channels play important signaling roles in the retina and olfactory neurons of other organisms, being activated by

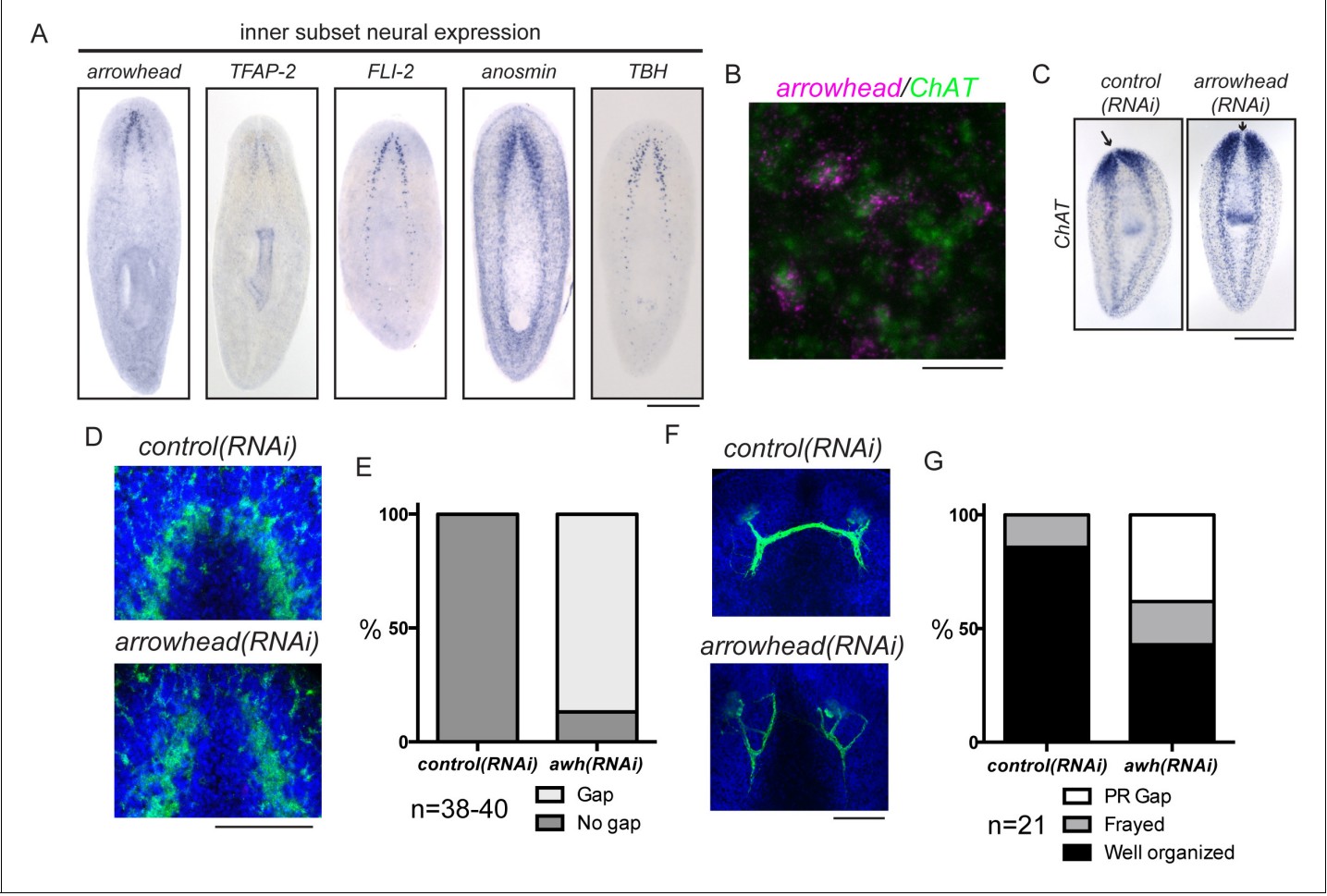

**Figure 5.** *arrowhead* is required for regeneration of medial structures in the planarian CNS. (**A**) Five upregulated genes were shown to be expressed medially in the CNS. These expression patterns were not overlapping (data not shown). (**B**) *arrowhead* is expressed in neurons, many of which are *ChAT*[+] (12/27). (**C**) *arrowhead(RNAi)* did not cause a small brain phenotype, but ISH with *ChAT* revealed a gap between the cephalic ganglia after regeneration. Arrows denote the location between cephalic ganglia in both samples. (**D**) The gap at the midline in *arrowhead(RNAi)* regenerates was visualized with an anti-synapsin antibody that stains the neuropil of the CNS. (**E**) 86.8% of *arrowhead(RNAi)* animals showed a gap at the midline, while this gap was not visible in any *control(RNAi)* animals (N = 38 and 40, respectively). (**F**) Photoreceptor neurons were visualized with the anti-arrestin (VC-1) antibody after regeneration of *control(RNAi)* and *arrowhead(RNAi)* animals. (**G**) Photoreceptor neurons were disorganized (19%) and often failed to cross the midline (38%) in *arrowhead(RNAi)* animals (N = 21). Scale bars: 500 μm (**A, C**), 20 μm (**B**), 100 μm (**D, F**).

The following figure supplement is available for figure 5:

**Figure supplement 1.** *arrowhead* and *slit* at the midline of the planarian brain.

cyclic nucleotides (cyclic AMP or cyclic GMP) downstream of sensory signaling (*Pifferi et al., 2006*; *Figure 6—figure supplement 1C,D*).

Because of the parallels between brain branches and chemosensory organs of other species, we sought to explore the roles of brain branch-expressed genes following regeneration. The 14 brain branch genes (*Figure 6A*) were expressed in a broad variety of patterns, ranging from single rows of cells (*hyp-c47858* and *cdc45-like*) to broad expression throughout the brain branches (*DL1* and *FLI-1*). We detected these transcripts in cells lateral to the neuropil of the cephalic ganglia, in or near cells expressing the classical brain branch marker *GluR1* (*Cebrià et al., 2002a*; *Figure 6—figure supplement 1A–B* and data not shown). To understand the function of these genes, we first examined whether the 14 brain branch genes affected regeneration of overall brain branch structure. As mentioned previously, *mblk(RNAi)* affects brain size during regeneration, but the other genes caused no

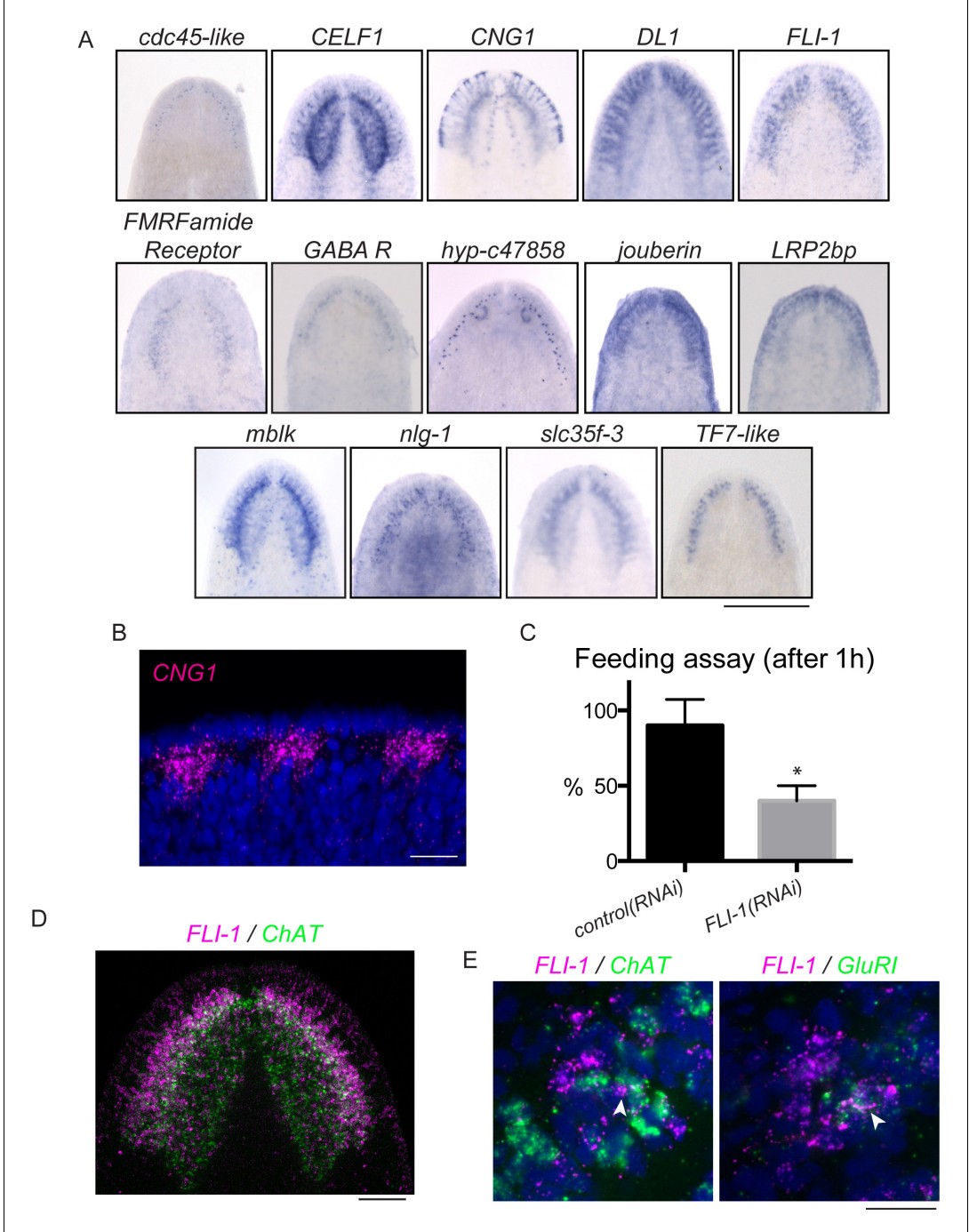

**Figure 6.** *FLI-1* is expressed in the brain branches and functions in reacquisition of chemosensory behavior during regeneration. (**A**) Fourteen upregulated genes were expressed in the brain branches of the planarian head. Expression patterns ranged from single rows of cells to broad expression in these structures. (**B**) *CNG1* is expressed in large, sub-epithelial regions of the planarian head margin. (**C**) Knockdown of *FLI-1* prior to regeneration caused a reduction of feeding in a maze-based experiment (shown in ***Figure 6—figure supplement 1E–H***). Only 40% of *FLI-1(RNAi)* animals ate in this assay, compared to 90% of *control(RNAi)* animals (9–10 animals each condition for each replicate, 3 replicates). Error bars depict SEM; results were significant with p=6.8 × 10$^{-5}$. (**D**) *FLI-1* is expressed in lateral structures of the planarian brain (here marked with *ChAT*). (**E**) Some *FLI-1*$^+$ cells express *ChAT* and *GluR1* (7/42 and 12/44 respectively). Scale bars: 500 μm (**A**), 20 μm (**B**, **E**), 100 μm (**D**).

The following figure supplement is available for figure 6:

**Figure supplement 1.** Assaying chemosensory function in the brain branches.

such defect. Brain branches, detected by *GluR1* ISH, were also present after knockdown of these 14 genes (data not shown).

To test whether brain branch-expressed genes promote chemosensory behavior, we repeated all RNAi treatments, this time assaying regenerated worms for their ability to find food in a simple maze (*Figure 6—figure supplement 1E–H*). The maze was designed so that planarians had to detect chemicals from their food (liver paste) and travel ~45 cm toward the food in order to eat. *FLI-1(RNAi)* animals consistently showed significantly reduced completion of the maze-based feeding assay, compared to *control(RNAi)* animals (*Figure 6C*). We confirmed the ability of *FLI-1(RNAi)* animals to move and to eat food (without traveling a maze) to rule out general defects in movement (*Figure 6— figure supplement 1I–J*, data not shown). *FLI-1* encodes an ETS-family transcription factor that is expressed in numerous cells of the planarian brain branches (*Figure 6A,D*). A minority of *FLI-1*[+] cells express *ChAT* or *GluR1* (*Figure 6E*, arrowheads), suggesting that the *FLI-1*[+] population likely represents a heterogeneous population of neural cells. Taken together, our findings: i) support a role for the planarian brain branches in chemosensory behavior and ii) suggest that FLI-1 is important for regeneration of some aspect of CNS structure or connectivity that restores chemosensory function during regeneration. Furthermore, our analysis of these genes reveals a previously underappreciated complexity to the brain branch structures, which must be reestablished during regeneration.

## Investigation of genes downregulated during regeneration reveals putative planarian glial cells

In addition to exploring the genes that are upregulated during regeneration as potential drivers of the process, we also investigated genes downregulated ($\geq$2x, p$\leq$0.01) after injury as potential inhibitors of regeneration. As proof of principle, we found that the planarian homolog of *activin*, an inhibitor of regeneration (*Roberts-Galbraith and Newmark, 2013*, *Gavino et al., 2013*), is downregulated during regeneration (*Supplementary file 1*). We identified 534 downregulated transcripts, with the number of downregulated genes increasing between 12 hr and 72 hr post-amputation (*Figure 7A*). We were interested in the types of cells that express these downregulated genes, so we cloned 70 genes and determined the expression patterns of 58 by ISH (*Supplementary file 3H*). The most frequent expression patterns for these genes were in the peripharyngeal secretory cells and the intestine (39% and 24%, respectively, *Figure 7—figure supplement 1A–C*), suggesting

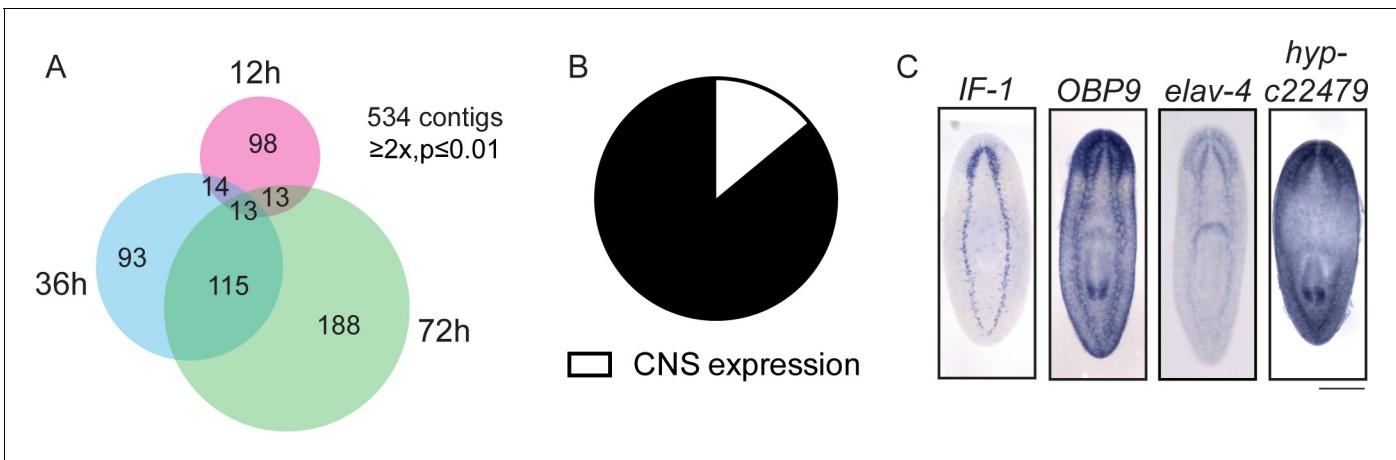

**Figure 7.** Expression profiling of genes downregulated during head regeneration. (**A**) Venn diagram showing the 534 genes significantly downregulated compared to cut controls ($\geq$2x downregulated, p$\leq$0.01). The number of downregulated genes increased over time. (**B**) Pie chart depicting the fraction of downregulated genes examined that are expressed in the CNS (8 of 58 with clear expression patterns). (**C**) ISH showing expression of several downregulated genes in the planarian CNS. *IF-1* is expressed in a pattern closely resembling the neuropil, while other markers (*OBP9*, *elav-4*, and *hyp-22479*) are typical of neural expression. Scale bars: 500 μm (**C**).

The following figure supplement is available for figure 7:

**Figure supplement 1.** Expression patterns of genes downregulated during head regeneration.

that cell loss or a change in the physiology of these organ systems occurs during the early stages of planarian head regeneration. We also detected a number of genes expressed in posterior regions (*Figure 7—figure supplement 1D*), an expected pattern as posterior genes likely need to be downregulated after amputation to permit rescaling of the body axis. Ultimately, we determined that 8/58 genes were expressed in the CNS (14%, *Figure 7B–C*). This proportion was about one third that of CNS expression patterns in the upregulated gene dataset.

Of the downregulated genes expressed in the planarian CNS, one expression pattern was distinct. Expression of an intermediate filament gene (*IF-1*) was detected in the CNS in a pattern similar to antibody markers of the neuropil and different than neural gene expression, which typically marks cell bodies around the CNS (*Figure 7C*). The expression pattern of *IF-1* led us to hypothesize that it is expressed in a non-neural subset of cells in the CNS. To test this possibility, we examined whether *IF-1* was coexpressed with markers of abundant neural types, including *PC-2* and *ChAT*. Indeed, *IF-1* was expressed in cells present in the neuropil that did not express neural markers (*Figure 8A*, *Figure 8—figure supplement 1A*, data not shown).

The nervous systems of many metazoans contain non-neural support cells that are broadly called glia. The existence of glial cells in planarians has been a matter of debate: electron microscopy analyses have been used to argue both for and against the presence of such cells (*Morita and Best, 1966*, *Lentz, 1967*, *Baguñà and Ballester, 1978*, *Golubev, 1988*). The identification of *IF-1*$^+$ glia is significant, especially given (i) the downregulation of *IF-1* as an injury response and (ii) the tendency of glial cells in other organisms to affect neural regeneration (positively or adversely depending on the situation – for review, see *Pekny et al., 2014*, and for more recent results see *Anderson et al. (2016)*). *IF-1* itself shares some similarity with intermediate filament genes expressed in neurons (vertebrate *neurofilament* and *C. elegans intermediate filament, ifa-1*) and glia (vertebrate *glial fibrillary acid protein*) (*Supplementary file 3G*). Thus, to determine whether the *IF-1*$^+$ cells of the planarian nervous system share similarities with glial cells in other organisms, we sought to identify genes coexpressed with *IF-1*. One frequent function of glial cells is to take up excess neurotransmitters or other chemicals that can be toxic to nearby neurons. For example, astrocytes take up glutamate and convert it to glutamine (*Schousboe et al., 1977*). Glutamate uptake occurs through excitatory amino acid transporters and conversion of glutamate to glutamine is catalyzed by glutamine synthetase (*Rothstein et al., 1994*, *Norenberg and Martinez-Hernandez, 1979*). We therefore investigated the expression patterns of planarian *slc1a-5 (solute carrier 1a-5 / EAAT)* and *GS-1 (glutamine synthetase-1)* and found that they were also expressed in a pattern similar to *IF-1* (*Figure 8B*). Furthermore, we detected *IF-1* coexpression with both *slc1a-5* and *GS-1* (*Figure 8C*). We also confirmed non-neural expression of *slc1a-5* using both *PC-2* and *ChAT* as markers of neurons (*Figure 8—figure supplement 1B*). Recently, a comprehensive characterization of solute carrier proteins in *S. mediterranea* was completed (*Vu et al., 2015*). We assayed the expression patterns of several additional solute carrier genes and determined that *IF-1*$^+$ cells express *slc1a-3* (an additional glutamate/neutral amino acid transporter), *slc2a-1* (a facilitative glutamate transporter), *slc6a-2* and *slc6a-8* (sodium and chloride-dependent sodium:neurotransmitter transporters), and *slc7a-8* (a cationic amino acid transporter) (*Figure 8C*, *Figure 8—figure supplement 1C*). Importantly, many of the genes coexpressed with *IF-1* are not differentially expressed (*slc1a-5* and *slc6a-2*) or are even slightly upregulated during regeneration (*slc2a-1*, *GS-1*, and *slc7a-8*). Taken together, these results indicate that *IF-1*$^+$ glial cells in planarians are similar to glial cells (and in particular astrocytes) in other organisms, in that they express a variety of transporters of the solute carrier family (*Cahoy et al., 2008*). Furthermore, because most markers coexpressed with *IF-1* are not reduced after amputation, we conclude that glial cells are not lost after injury, but instead alter their gene expression.

We also sought an unbiased way of identifying genes coexpressed with *IF-1*, particularly additional genes downregulated during head regeneration. Reasoning that genes that behave like *IF-1* during regeneration might meet both criteria, we turned to our clustering analysis, and focused on module 2, a cluster of 1520 genes (including *IF-1*) that are downregulated by 12 hr post-amputation and even more significantly downregulated at 36 hr and 72 hr (*Figure 8—figure supplement 1D–E*). As a pilot study, we cloned 48 genes from this module and examined their expression patterns. The majority of these genes showed expression patterns irrelevant to our glial studies. However, two transcripts were expressed in *IF-1*$^+$cells (*Figure 8D–E*). *jagged-like* encodes a protein with weak similarity to Jagged that contains a signal peptide, two EGF domains, and a transmembrane domain. We also identified a gene that encodes a signal peptide-containing protein; we named this

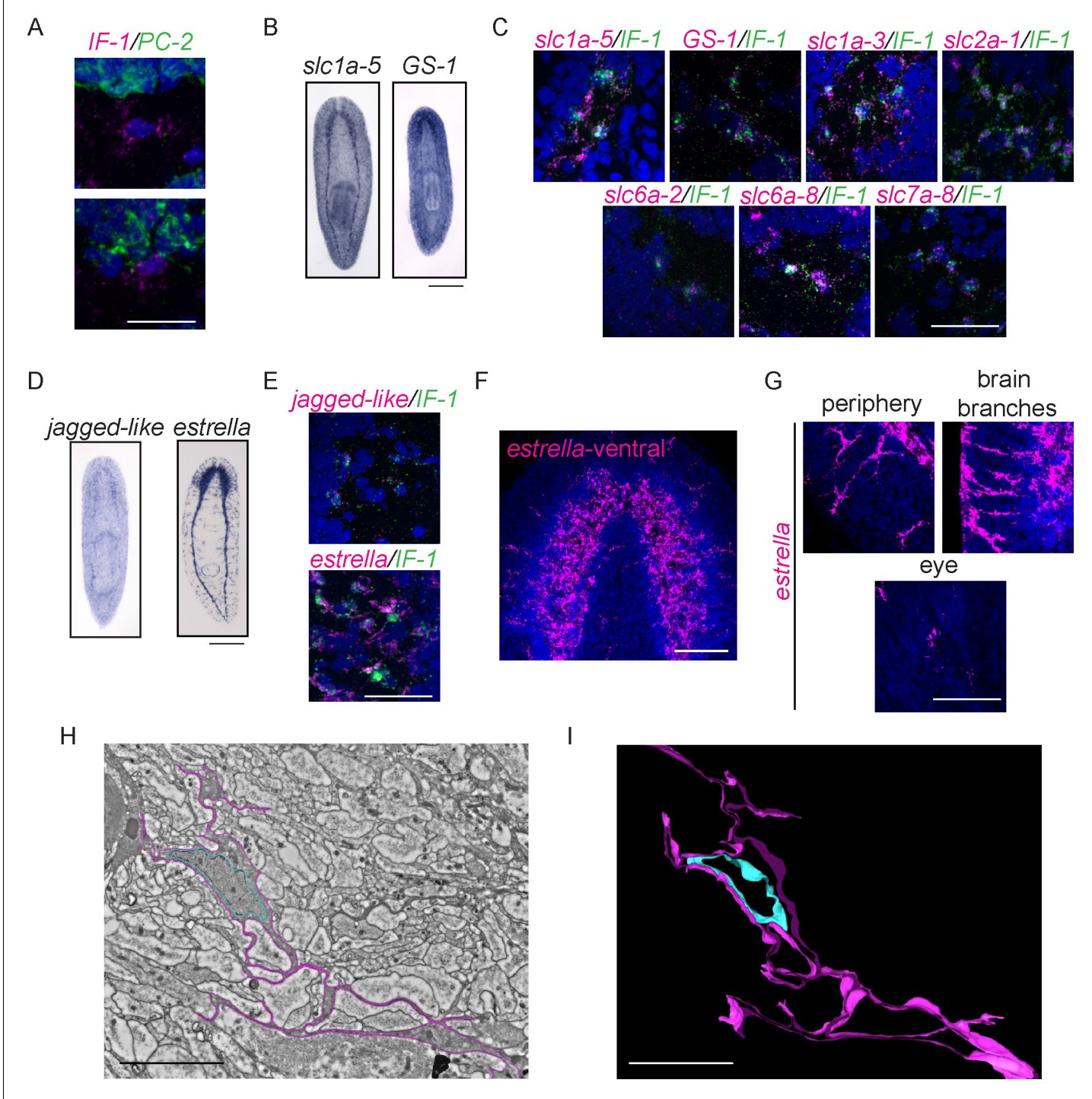

**Figure 8.** *Intermediate Filament (IF-1)* gene expression marks planarian glial cells. (**A**) FISH was used to show that *IF-1* is not coexpressed with *prohormone convertase-2 (PC-2*, 0/25 *IF-1*[+] cells were *PC-2*[+]). (**B**) *slc1a-5* and *glutamine synthetase 1 (GS-1)*, genes required for glutamate uptake and conversion to glutamine are also expressed in an *IF-1*-like neuropil pattern. (**C**) *IF-1* is co-expressed with *slc1a-5* and *GS-1* (46/62 *IF-1*[+] cells were *slc1a-5*[+] and 56/56 *IF-1*[+] cells were *GS-1*[+]). *IF-1*[+]cells also express the solute carrier genes *slc1a-3* (33/33), *slc2a-1* (76/76), *slc6a-2* (31/35), *slc6a-8* (48/48) and *slc7a-8* (63/63). Overlap was near perfect for some genes (all *slc7a-8*[+] cells express *IF-1*), but other genes are more widely expressed (only 48/347 *slc6a-8*[+] cells express *IF-1*). (**D**) Two genes from module 2 (a downregulated gene cluster) are also expressed in a neuropil-like pattern. *jagged-like* is expressed in puncta and a novel gene, *estrella,* is expressed strongly in the neuropil and in cells around the planarian body. (**E**) *IF-1*[+] cells express *jagged-like* and *estrella* (48/48 and 61/61, respectively). (**F–G**) FISH with the *estrella* probe marks cells with extensive cytoplasmic projections in the neuropil, in the periphery of the animal body, near the brain branches, and in the eye, suggesting that *estrella*[+] cells surround many components of the

*Figure 8 continued on next page*

*Figure 8 continued*

planarian nervous system. (H) An electron micrograph of a putative planarian glial cell in the neuropil of the ventral nerve cord. The cell membrane (determined by images acquired at higher magnification) is marked in magenta and the nuclear envelope is shown in cyan. The cytoplasm is more electron dense than that in surrounding axons. The nuclear morphology is elongated, with heterochromatin at the nuclear periphery; these features are not present in neurons visualized by EM. (I) A partial reconstruction of the cell in (H) through ten serial sections. The cell membrane and nuclear envelope are colored as above. Scale bars: 500 μm (B, D), 20 μm (A, C, E, G), 100 μm (F), 5 μm (H–I).

The following figure supplement is available for figure 8:

**Figure supplement 1.** Characterization of planarian glial cells.

novel gene *estrella* ('star' in Catalan) based on its expression pattern and the origin of our *S. mediterranea* strain in Barcelona. The expression pattern of *estrella* highlights the extensive processes of glial cells and their wide distribution in the neuropil, the peripheral nervous system, near the brain branches, and near photoreceptors (*Figure 8F–G*; *Cowles et al., 2014*). Together, our results indicate that *IF-1*$^+$ glial cells respond to injury by downregulating a variety of genes (including genes that encode potential signaling molecules) over the course of several days.

Finally, we used electron microscopy to confirm the existence of glial cells in the planarian neuropil. Indeed, we detected cells in the neuropil that had numerous fine cellular processes that intermingled with axons (*Figure 8H–I*, *Video 3–4*). Planarian glial cells were also distinct from nearby neurons in that they had electron-dense cytoplasm and elongated nuclei with abundant peripheral heterochromatin (*Figure 8H*). Throughout the neuropil, where neuronal processes are a dominant feature, electron-dense projections were also common (*Video 5*), suggesting extensive projection of glial cells throughout the neuropil of the planarian nervous system. Though we have not yet shown that the cells identified by electron microscopy are *IF-1*$^+$, the morphology seen by electron microscopy is consistent with the cell shapes revealed by *estrella* gene expression (*Figure 8F–G*). We therefore conclude that both our gene expression data and electron microscopic analysis support the existence of glia in the planarian CNS.

## Discussion

After injury, organisms execute a suite of molecular and cellular responses. On one end of the regenerative spectrum, the injury response can minimize damage and cordon off injured tissue. In organisms like planarians, however, an exquisite set of events results in the precise and reproducible replacement of missing tissues and complete restoration of structure and function. While a general picture of the wound response has emerged from planarians and other model organisms (*Wenemoser et al., 2012*), we do

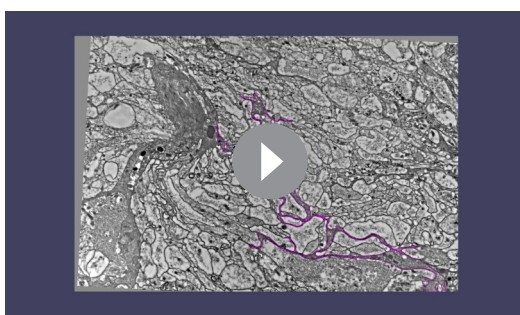

**Video 3.** Electron micrographs of the glial cell shown in *Figure 8H* through serial sections.

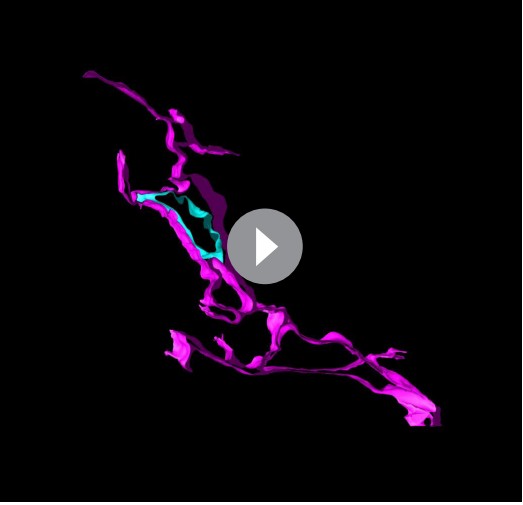

**Video 4.** Rotation of the reconstruction shown in *Figure 8I*.

not yet understand how this wound response can be faithfully channeled into a regenerative response. Studies of planarians and other regenerative model organisms have begun to provide an understanding of regeneration in broad strokes. In planarians, stem cell activity drives cell replacement and a reestablished coordinate system of axial polarity cues redirects the planarian body plan. Still, many basic questions remain, particularly in understanding how stem cell activity and polarity intersect with other cellular events to specifically produce missing organs like the CNS.

## An instruction manual for brain regeneration: novel signaling factors

To better understand regeneration, it will be important to identify the signals that initiate and direct the process. Some signals, from small molecules to proteins, have been identified in other regenerative model organisms (e.g., Wnt3 in *Hydra*, $H_2O_2$ and cysteinyl leukotriene in zebrafish, a MARCKS-like protein in axolotl, and newt anterior gradient protein; *Chera et al., 2009*, *Kyritsis et al., 2012*, *Niethammer et al., 2009*, *Kumar et al., 2007*, *Sugiura et al., 2016*). Surprisingly, the signals that promote regeneration in planarians remain mysterious. Notwithstanding a host of signaling molecules known to establish polarity in the regenerating worm, only a few signaling molecules or their receptors have been determined to regulate regeneration, including opposing Follistatin and Activin signals and EGF- and FGF-receptors that regulate various aspects of homeostasis and regeneration (*Roberts-Galbraith and Newmark, 2013*, *Gavino et al., 2013*, *Fraguas et al., 2011*, *Ogawa et al., 2002*, *Wagner et al., 2012*, *Lei et al., 2016*). Importantly, though planarian neoblasts have been hypothesized to integrate a wide variety of signals to choose between daughter cell fates (*Reddien, 2013*), whether neoblasts express the diversity of receptors required to integrate numerous signals remains unknown. Neoblasts do express FGF-receptors that are important for tissue maintenance (*Ogawa et al., 2002*, *Wagner et al., 2012*), but no ligands for these receptors have been identified through homology searches. Thus, further exploring the signals required for regeneration, as well as the cellular origin(s) and receiver(s) of each signal will greatly enhance our understanding of how diverse cellular events (i.e., proliferation, migration, organization, differentiation) are triggered and coordinated during the regenerative response. One goal of our screen was to identify putative signals that influence regeneration, generally, or for the CNS in particular. To that end, we identified several putative signaling molecules encoded by *CRELD*, *F-spondin*, and *LDLRR* family genes that each encode a secreted or transmembrane protein that could function in a signaling capacity.

*CRELD* genes are conserved among metazoans but are not very well characterized. Two *CRELD* homologs exist in mammals. *CRELD1* serves an essential function during mouse heart development and encodes a protein reported to have two transmembrane domains and a cytoplasmic EGF-like domain (*Mass et al., 2014*). Conversely, *CRELD2* encodes a secreted protein (*Oh-hashi et al., 2011*). In mice, *CRELD1* is expressed in the CNS in addition to its expression in the heart, and the *Drosophila CRELD* homolog (*CG11377*) is expressed across multiple tissues, including the nervous system (*Mass et al., 2014*; *Brown et al., 2014*). Planarian *CRELD* is expressed in diverse cell types (*Wurtzel et al., 2015*), including stem cell progeny (*Labbé et al., 2012*), and is enriched in the brain and head. *Smed-CRELD* has two predicted isoforms (*Figure 3—figure supplement 1E*; *Brandl et al., 2016*) that encode a transmembrane protein and a secreted protein, each with extracellular EGF-like domains. Further studies will be required to determine how *CRELD* exerts its function in the regenerating head and to what extent this function might be conserved.

Other signaling molecules identified in our study include homologs of matrix-associated proteins. F-spondin proteins are conserved among metazoans, localize to ECM-rich areas, and play roles in cell adhesion, including promoting axon extension from the neural floor plate (*Klar et al., 1992*; *Umemiya et al., 1997*; *Burstyn-Cohen et al., 1998*; *Burstyn-Cohen et al., 1999*; *Woo et al., 2008*). Despite their names, planarian LDLRR proteins are most closely related to basement membrane-specific Heparan Sulfate Proteoglycans (HSPGs) in other metazoans. In other organisms, HSPGs are also ECM components and modulate signaling of many ligands, including Wnt, Hedgehog, and FGF (*Tsuda et al., 1999*; *Reichsman et al., 1996*; *The et al., 1999*; *Kiefer et al., 1990*). F-spondin and the LDLRR proteins are predicted to be secreted molecules and could promote regeneration by serving as signals themselves, by binding to and changing the activity of other signaling molecules, and/or by contributing to an unknown regenerative function of the planarian ECM.

Importantly, *F-spondin*, *LDLRR-1*, and *-3* are each expressed in differentiated, non-muscle cells in the parenchyma, suggesting that these uncharacterized cells perform some signaling function

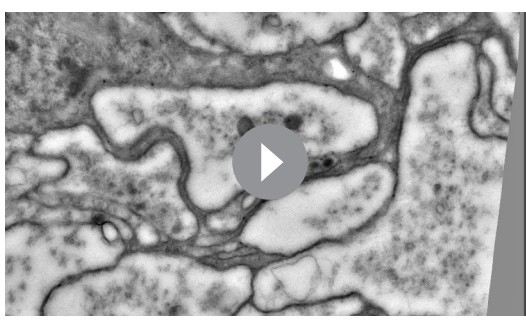

**Video 5.** Close-up from serial sections showing electron-dense projections of glial cells among axons in the neuropil.

important for proper regeneration. Cells expressing *LDLRR-3* and *F-Spondin* clustered with 'gut' cells in a recent transcriptomic analysis of single cells (*Wurtzel et al., 2015*), but the overall expression patterns of these genes are not reflective of the intestine. Thus, further work will be required to identify and characterize these cells. Planarian stem cells are present throughout the parenchyma and are surrounded by other parenchymal cell types, one of which has been termed 'fixed parenchyma cells' in the electron microscopy literature (*Pedersen, 1961*). We have yet to determine what kind of cell-cell interactions or local signaling events in the parenchyma influence the activity of planarian stem cells, either in homeostasis or in regeneration. The parenchymal signaling cells and molecules revealed by our screen might be a valuable entry point in determining whether or how neighboring cells promote the maintenance, cycling, differentiation, or migration of planarian neoblasts.

Another area of investigation concerns whether the CNS influences regeneration by playing a signaling role, as it does in other organisms (*Kumar and Brockes, 2012*). A recent single-cell analysis suggested that neurons do not respond to injury with gene expression changes (*Wurtzel et al., 2015*). However, these studies were performed on worm fragments that were missing their cephalic ganglia, and only addressed gene expression changes in the first 12 hr after amputation. We identified one gene upregulated later in regeneration, *arrowhead*, that is required at the midline of the cephalic ganglia for axon organization across the anterior commissure. This non-cell-autonomous role likely involves signaling from *arrowhead⁺* neurons at this location to organize the projections of other cells, including the photoreceptor neurons. Our preliminary finding that some *arrowhead⁺* cells express *slit* offers one clue about how this might occur, but a deeper investigation of Arrowhead targets will distinguish between this and other possibilities. Thus, at a minimum, some neurons of the CNS likely influence the organization of one another during regeneration. Further work will be required to identify the molecules that govern this influence and to study other contributions of the CNS to the signaling environment required for a proper regenerative response.

## CNS regeneration: a cellular parts list

One of the crucial events required for regeneration of nervous tissue is coordinated neurogenesis. In the planarian field, a major question is how neoblasts are directed to specific cell fates, including neurons, and how the appropriate number and mixture of cells are reestablished. In planarians, some work has been done to identify a handful of factors that regulate commitment to certain neural subtypes, including *coe, lhx1/5–1*, and others (*Currie and Pearson, 2013*; *Cowles et al., 2013*; *Scimone et al., 2014*). Additionally, a new category of specialized neoblast has recently been proposed to contribute to neural fates (*Molinaro and Pearson, 2016*). However, we lack a fundamental understanding of the basic neurogenesis program(s) in planarians. It is not yet clear whether neoblasts commit to specific neural fates in one step or through a series of binary choices. Additionally, we do not know how or whether these fate choices are reoriented in response to injury.

In our screen, we expected to identify factors that could play roles in neural cell fate decisions. We identified a transcription factor, *soxB2-2*, required for full expression of several neural progenitor genes. Importantly, *soxB2-2(RNAi)* animals display behavioral phenotypes (decreased sensitivity to vibrations) and regenerate brains smaller than those resulting from many other perturbations. These observations suggest that *soxB2-2* likely regulates gene expression even more broadly than we have so far observed. The homolog of *soxB2-2* is expressed during embryonic development of *Schmidtea polychroa* (*Monjo and Romero, 2015*). Related Sox family members *sox14* and *sox21* play roles in the neurobiology of diverse animal species (*Hargrave et al., 2000*, *Kirilly et al., 2009*, *Sandberg et al., 2005*, *Richards and Rentzsch, 2015*). In particular, *sox21* is upregulated during spinal cord regeneration in *Xenopus* (*Lee-Liu et al., 2014*). It will be important to determine whether roles for *soxB2-2* in neurogenesis and regeneration are indeed conserved.

We also expanded the known roles for planarian *runt* in brain regeneration. *runt* is expressed in response to different injuries in planarians (*Wenemoser et al., 2012*, *Sandmann et al., 2011*) and modulates factors, including *sp6-9* and *ovo,* that are essential for eye development (*Figure 3F*; *Sandmann et al., 2011*, *Wenemoser et al., 2012*, *Lapan and Reddien, 2011*, *Lapan and Reddien, 2012*). In addition to its role in the eye, *runt* is important for proper expression of *ap2* and *klf* (*Figure 3F*; *Wenemoser et al., 2012*); *ap2* drives respecification of medial *TrpA+* neurons, and *klf* is important for regeneration of *cintillo$^+$* cells in the brain branches (*Wenemoser et al., 2012*, *Scimone et al., 2014*). Thus, *runt* influences neural progenitors to contribute to multiple neural types across different regions of the brain.

Neither *soxB2-2* nor *runt* appears to be required globally for neurogenesis, but each controls a distinct array of fates and is required for optimal brain regeneration, suggesting that these two genes' products promote distinct neurogenesis pathways. Both *soxB2-2* and *runt* are upregulated during regeneration, implying that they are activated in neoblasts or early progenitors to skew the balance of specialized progeny and promote neural fates needed for rebuilding the brain. Future studies could use *soxB2-2* and *runt* as starting points for defining neurogenesis pathways; by exploring both upstream regulators and targets that control downstream progenitor and neural fates, these middle-out studies could define differentiation trajectories from neoblast to neuron.

Dozens of neural cell types in the planarian CNS can be defined by the expression patterns of neuropeptide and neurotransmitter biosynthesis genes (*Nishimura et al., 2007a*, *2007b*, *2008a*, *2010*, *2008b*, *Collins et al., 2010*). Most of these neural subtypes arise during regeneration through mechanisms that have not yet been uncovered. Exceptions include the specification of serotonergic neurons by *lhx1/5–1* and *pitx* (*Currie and Pearson, 2013*, *Marz et al., 2013*) and the specification of *dopamine ß-hydroxylase$^+$* neurons by *pax3/7* (*Scimone et al., 2014*). In our study, RNAi knockdown of *runt* failed to significantly influence any of the specific markers of differentiated neurons that we tested in this analysis (e.g., *ChAT* to mark cholinergic neurons). Although knockdown of *soxB2-2* decreased several markers, only *tyrosine hydroxylase (TH)* dropped past the threshold of statistical significance (*Figure 3C*). That RNAi would affect presumptive neural progenitors without a detectable effect on a neural marker could simply be due to a lack of tools required to delineate the affected neural cell types, or due to the sensitivity of the assay. However, other transcription factors identified in this screen as having no brain-size phenotype (e.g., *glass*) could play more subtle roles in specification or function of neural subtypes. *FLI-1* expression in neoblasts could shift them to certain fates important for brain-branch architecture or wiring. In contrast, transcription factors like *arrowhead* and *mblk* are each expressed predominantly in differentiated cell types (*Labbé et al., 2012*), and likely function later in differentiation or in mature neurons to affect maintenance, connectivity, or function. Along these lines, a homolog of *mblk* in *C. elegans* is important for neurite pruning (*Kage et al., 2005*).

Surprisingly, though we anticipated that *hesl-3* would affect gene expression in neural progenitors (*Cowles et al., 2013*), we found that the clearest impact of *hesl-3(RNAi)* is on polarity genes like *sFRP* and *foxD*. *hesl-3* expression is highest in neoblasts (*Cowles et al., 2013*, *Labbé et al., 2012*), though expression in the CNS is also evident (*Figure 3G*; *Cowles et al., 2013*). However, no expression at the anterior tip is visible. One straightforward model for the function of *hesl-3* would be that it is required for specification of muscle cells that eventually contribute to the anterior pole. Another possibility is that *hesl-3$^+$* cells contribute to regeneration of the brain and that the regenerating brain is important for resetting or maintaining anterior polarity. Neural tissue is critical for regeneration in a variety of other model organisms (*Kumar and Brockes, 2012*), but we have not been able to determine whether it is so in planarians. Future work will be required to identify targets of *hesl-3* and to determine whether its roles in head patterning and brain regeneration occur through an influence on neurogenesis, polarity, or both.

## The "glue" to hold it together?

Glia exist across animal phyla and play diverse, crucial roles in supporting the differentiation, survival, and function of neurons. We investigated genes downregulated during planarian regeneration and discovered a marker (*Intermediate Filament-1*) that is expressed in non-neural cells that have several hallmarks of glial identity. First, *IF-1$^+$* cells express a range of solute carrier family genes, including genes that are important for reuptake of glutamate; this function is carried out by astrocytes in the mammalian nervous system (*Schousboe et al., 1977*, *Norenberg and Martinez-*

Hernandez, 1979). Second, IF-1[+] cells are present in the neuropil and have extensive cytoplasmic projections, positioning them ideally to support axonal or synaptic activity of neurons in the CNS. Third, IF-1[+] cells respond to injury by changing their gene expression (downregulating *IF-1*, *jagged-like*, and *estrella*), much like astrocytes in mammals (for reviews, Sofroniew, 2009, Pekny and Nilsson, 2005).

The discovery of glia in the planarian CNS raises many interesting possibilities for future study. Glia are an abundant, diverse, and essential population of cells in the mammalian nervous system, so functional regeneration of human neural tissue would require regeneration of glia alongside regeneration of neurons themselves. Thus, it will be informative to understand how glia repopulate the regenerating planarian CNS and in particular how glial and neural regeneration are coordinated. Additionally, activation of astrocytes in the mammalian nervous system protects tissue after injury but also contributes to glial scarring that can promote or prohibit axonal regrowth or functional recovery (Pekny et al., 2014, Anderson et al., 2016). Though it is not yet clear whether the planarian glial injury-response program is important for driving or permitting regeneration, contrasting the planarian glial response with that of glia in non-regenerating organisms might shed light on how glial activation can be harnessed for purely regeneration-promoting functions. For example, planarian *IF-1* is downregulated after injury, while mammalian *GFAP* is upregulated during astrocyte activation (Condorelli et al., 1990). Finally, RNAi of the glial genes identified in this study has so far failed to reveal phenotypes (data not shown). This is not surprising, because the downregulation of *IF-1* and other glial genes during regeneration suggests that gain-of-function treatments might be more likely to perturb regeneration. More nuanced assays and identification of genes required for glial specification or maintenance will be required to assess whether and how planarian glia support neuronal biology and behavior in planarians.

## Conclusion

Planarians regenerate their central nervous system *de novo*, prodigiously replacing neurons and reconnecting neural networks to restore function in only a week. As such, they provide a simple, in vivo model for understanding neural regeneration. In this study, we used a transcriptomic approach to guide an unbiased functional screen. Through RNAi studies, we identified several factors that influence reestablishment of cell types, reconnection of a major commissure, and restoration of chemosensory behavior. Genes identified in our study also mark new cell types, including planarian glia, putative chemosensory cells, and previously unknown parenchymal cells that promote regeneration. In sum, we have advanced our understanding of diverse aspects of planarian neurobiology and neural regeneration; furthermore, we have laid the foundation for a wide range of studies geared toward understanding success and failure in neural regeneration.

# Materials and methods

## Planarian husbandry

Planarians from an asexual, clonal line (*Schmidtea mediterranea*, CIW4; Sánchez Alvarado et al., 2002) were maintained at 21°C in ultrapure water with 0.5 g/L Instant Ocean salts (Spectrum Brands, Blacksburg, VA). Regeneration experiments were completed in the presence of 50 μg/mL gentamicin (Gemini Bio-Products, West Sacramento, CA). Animals were stored in unsealed Ziploc containers or 60 mm petri dishes. For regeneration experiments, animals were amputated using sterile scalpels. Animals were between 2–5 mm for all experiments unless otherwise noted and were starved for at least one week prior to use. For stem cell ablation, animals were irradiated with 60 gray at 150 kV, 5 mA on the top shelf of a Gammacell-220 with a cobalt-60 source (Nordion, Ottawa, ON, Canada).

## Gene expression analysis

Large worms (~1 cm) were used for transcriptomic studies. Animals were amputated post-pharyngeally and were allowed to regenerate for 12, 36, or 72 hr. Whole animals and animals killed immediately after amputation were used for two separate controls. Either 10 whole animals or 30 tail pieces were used per condition, in triplicate. Planarians were killed in TRIzol reagent (Thermo Fisher Scientific, Waltham, MA) and samples were frozen on dry ice and stored at −80°C until RNA

purification was completed as per the manufacturer's protocol. After purification, RNA was DNase-treated (Promega, Madison, WI) and column purified (Zymo Research, Irvine, CA). RNA samples were submitted to the W. M. Keck Center for Comparative and Functional Genomics (UIUC) for quality analysis using a bioanalyzer (Agilent, La Jolla, CA) and subsequent library construction and sequencing (Illumina, San Diego, CA). Sequencing depth was 13.5–40.1 million reads (100 nt) per sample, for a total of 325.5 million reads across all conditions. All Illumina sequencing was deposited in the Sequence Reads Archive as study PRJNA319973. Initially, reads were mapped via RNA-Seq (*Marioni et al., 2008*) in the CLC genomics workbench (QIAgen, Hilden, Germany) to a previously generated transcriptome (*Rouhana et al., 2012*; available online at www.ideals.illinois.edu/handle/2142/28689 and within the PlanMine database (*Brandl et al., 2016*), planmine.mpi-cbg.de). RPKM (reads per kilobase of transcript per million mapped reads) values were calculated and compared using the CLC suite, with p values calculated for each pair of samples using two-sided, unpaired T-tests with Bonferroni correction (*Supplementary file 1*). From this analysis, contigs ≥two-fold upregulated or downregulated (p≤0.01) were pursued for initial functional analysis.

In parallel, raw reads for each contig were input into R 3.1.1 (*R Core Team, 2013*) for data pre-processing and statistical analysis using packages from Bioconductor (*Gentleman et al., 2004*) as below. After trimmed mean of M-values normalization (*Robinson and Oshlack, 2010*) and filtering of low abundance transcripts (<0.5 counts per million were discarded), the remaining 31,862 (of 55,949) contigs were analyzed using edgeR v 3.6.8 (*Robinson et al., 2010*). A one-way ANOVA was calculated across the five groups for each contig using a quasi-likelihood negative binomial general-ized log-linear model (*Lund et al., 2012*) with robust estimation of the sample variances. Multiple hypothesis test correction was completed using the False Discovery Rate (FDR) method (*Benjamini and Hochberg, 1995*). To examine global gene expression patterns across the dataset, we selected 9,850 contigs with one-way ANOVA FDR p-value ≤0.1 for weighted gene correlation network analysis (WGCNA) (*Zhang and Horvath, 2005*, *Langfelder and Horvath, 2008*). We per-formed WGCNA (v 1.41–1) using the default values of the blockwiseModules() function except for: soft thresholding power beta = 2-, TOMType = "signed," a minimum module size of 20, and merg-ing similar modules at 0.15; this approach yielded 40 modules, each with genes that share similar expression dynamics across the time course. An eigengene value was computed for each sample from the first principal component score of the expression values of the contigs in the module.

## Molecular biology methods

For each gene of interest, a 300–1000 bp fragment was cloned from cDNA into pJC53.2, a vector designed to allow TA-cloning and subsequent production of riboprobes or dsRNA (*Collins et al., 2010*), using standard molecular biological methods. Riboprobes and dsRNA were synthesized as previously described (*Collins et al., 2010*, *Rouhana et al., 2013*). For all RNAi experiments, dsRNA matching *GFP* or bacterial *ccdB* genes was used for negative controls. Riboprobes were synthesized with digoxigenin-12-UTP (Roche, Basel, Switzerland) or dinitrophenol-11-UTP (Perkin Elmer, Waltham, MA). Quantitative PCR experiments were performed as previously described (*Miller and Newmark, 2012*). Briefly, RNA was purified as described above and cDNA was synthe-sized using an iScript kit (Bio-Rad, Hercules, CA). qPCR was performed using GoTaq Mastermix (Promega) on a StepOnePlus real-time PCR machine and software (Applied Biosystems, Foster City, CA). ß-tubulin was used as a normalization control and qPCR primers are listed in *Supplementary file 3E*. Initial qPCR experiments were performed 2–4 times per gene knockdown; validation qPCR experiments (*Figure 3*) were performed in biological triplicate and technical tripli-cate, except *soxB2-2(RNAi)* and control, for which we had 6 biological replicates and technical tripli-cates. p values were calculated by t test in Prism (GraphPad, La Jolla, CA).

Sequence analysis was performed to detect protein domains (pFam 29.0; *Finn et al., 2016*), trans-membrane domains (TMHMM 2.0; *Krogh et al., 2001*), and signal peptides (SignalP 4.1; *Petersen et al., 2011*). Sequences for genes identified in this analysis have been deposited at Genbank.

## in situ hybridization and immunofluorescence

Colorimetric in situ hybridization (ISH) experiments were done as per *Pearson et al. (2009)*, but with the following changes described in *King and Newmark (2013)*: the reduction step was eliminated;

animals were bleached in a formamide-containing solution for 2.5–3 hr under light; and proteinase K was used at a concentration of 5 µg/mL (for regenerating worms) or 10 µg/mL (whole animals). For colorimetric ISH, Digoxigenin-containing probes were used in combination with an anti-DIG-AP antibody (1:2000, Roche). Fluorescent in situ hybridization (FISH) was performed as above, but with the following changes: digoxigenin- and/or dinitrophenol-containing probes were used; riboprobes were detected by anti-digoxigenin-POD (1:2000, Roche) or anti-dinitrophenol-HRP (1:300, Perkin Elmer); the blocking solution was TNTx (100 mM Tris pH 7.5, 150 mM NaCl, 0.3% Triton X-100) with 5% horse serum and 0.5% Roche western blocking reagent (Roche); and TNTx was also used for post-antibody washes. For FISH, tyramide conjugates were synthesized and signal amplification reactions were performed as described (*King and Newmark, 2013*).

Immunofluorescence experiments were performed as described (*Roberts-Galbraith and Newmark, 2013*). The following primary antibodies were used: anti-phospho-histone H3 (1:5000, S10; Cell Signaling Technology, Danvers, MA), anti-synapsin (1:100, 3C11 concentrate; Developmental Studies Hybridoma Bank, Iowa City, IA), and anti-arrestin (1:10000, VC1; kindly provided by Kiyokazu Agata). Secondary antibodies (Molecular Probes, Eugene, OR) were used at a dilution of 1:800 (goat anti-mouse Alexa Fluor 488) or 1:2500 (goat anti-rabbit Alexa Fluor 568).

## RNA interference and behavioral assays

Animals (10–12) were placed in a 60 mm petri dish in salts and were fed 3 µg dsRNA in 30 µL of a liver puree:salts mixture (3:1, tinted with green food coloring). RNAi experiments are diagrammed in *Figure 2A* and *Figure 6—figure supplement 1H*. Briefly, animals were fed three times over the course of 10–11 days, amputated prepharyngeally 5 days after the last feeding, and allowed to regenerate for 6 hr (*Figure 4—figure supplement 1B–C*), 6 days (for all other in situ or immunofluorescence experiments), or 10 days (behavioral assays). For all experiments, animals were observed for obvious behavioral or physiological defects before they were fixed or otherwise assayed. To assay blastema size, animals were fixed after regeneration for imaging.

Mazes were constructed at the Life Sciences Machine Shop (University of Illinois) out of clear and black acrylic sheets according to the measurements shown (*Figure 6—figure supplement 1E–F*). Prior to a behavioral experiment, we added 70 mL salts to each maze and pipetted 60 µL of a liver puree:salts mixture (3:1, tinted green) into the far side of the maze as shown (*Figure 6—figure supplement 1G*). The mazes were incubated at room temperature for fifteen minutes. RNAi-treated, regenerated animals (10–12 per replicate) were placed in the maze in the position shown (*Figure 6—figure supplement 1G*). Mazes were covered to allow complete darkness and animals were permitted 1 hr to complete the maze and eat. Animals that had eaten were scored based on the presence of green in their intestines. Experiments were performed in biological triplicate and differences between the samples were evaluated using a Chi-squared test. In a separate experiment, control and *FLI-1(RNAi)* animals were added to the center of 60 mm petri dishes on a sheet with concentric circles (the smallest of which was 1.5 cm in diameter). After 15 min, animals were scored for their ability to move beyond the starting circle. A Chi-squared test was also used to evaluate the difference between these samples.

## Imaging

We used a Leica M205A stereomicroscope to image animals live or after colorimetric ISH. Images were captured using LAS 3.6.0 software (Leica, Wetzlar, Germany). We quantified brain area and blastema size in ImageJ (*Schneider et al., 2012*). Areas were averaged across a condition and were compared to control using a one-way ANOVA. Fluorescence images (immunofluorescence and FISH) were captured using a Zeiss LSM 710 confocal microscope and either a 20X (Plan-Apochromat 206/0.8) or a 63X objective (Plan-Apochromat 636/1.4). Zen software (Carl Zeiss, Oberkochen, Germany) was used for these experiments.

## Electron microscopy

We fixed and processed planarians for transmission electron microscopy as described in *Brubacher et al., 2014*. Serial 70 nm sections were cut on a Leica Ultracut S ultramicrotome, and collected on formvar- and carbon-coated copper slot grids (Ted Pella, Redding, CA) and stained with lead citrate for 2 min at room temperature (*Venable and Coggeshall, 1965*). We imaged the

trunk ventral nerve cord using a Hitachi H-7000 STEM at 75 kV, equipped with an AMT 1600 M CCD camera (Hitachi, Tokyo, Japan). IMOD software version 4.3.7 (*Kremer et al., 1996*) was used to stack and align images, and to segment and mesh digital models. Image processing and animations were conducted with Adobe Photoshop CS5 software (Adobe Systems, San Jose, CA).

## Acknowledgements

The authors would like to thank the members of the Newmark laboratory, past and present, for their constructive feedback over the course of this project. We also thank Amanda Brock, Melanie Issigonis, Umair Khan, Tania Rozario, and Rachel Smith-Bolton for critical reading of this manuscript. We thank Alvaro Hernandez and the staff of the WM Keck Center for Comparative and Functional Genomics (UIUC) for help with the design and execution of our RNA sequencing experiments. We thank Jenny Drnevich at High Performance Biological Computing (UIUC) for clustering and further analysis of our RNA sequencing data. We thank Scott Baker and Jared Bear in the Life Sciences Machine Shop (UIUC) for construction of mazes used in behavioral experiments. We also thank Andrew Belmont (UIUC) for allowing us to use his ultramicrotome for sectioning. RHR-G was a fellow of the Jane Coffin Childs Memorial Fund for Medical Research. JLB's participation was partially supported by Faculty Research Grants from Canadian Mennonite University. PAN is an investigator of the Howard Hughes Medical Institute.

## Additional information

### Funding

| Funder | Grant reference number | Author |
|---|---|---|
| Howard Hughes Medical Institute | | Phillip A Newmark |
| Jane Coffin Childs Memorial Fund for Medical Research | | Rachel H Roberts-Galbraith |
| Canadian Mennonite University | Faculty Research Grant | John L Brubacher |

The funders had no role in study design, data collection and interpretation, or the decision to submit the work for publication.

### Author contributions

RHR-G, JLB, Conception and design, Acquisition of data, Analysis and interpretation of data, Drafting or revising the article; PAN, Conception and design, Analysis and interpretation of data, Drafting or revising the article

### Author ORCIDs

Rachel H Roberts-Galbraith, http://orcid.org/0000-0002-2682-2366
John L Brubacher, http://orcid.org/0000-0002-2346-9245
Phillip A Newmark, http://orcid.org/0000-0003-0793-022X

## Additional files

### Supplementary files

• Supplementary file 1. All gene expression data from this study. Read mapping and expression levels are given for each contig in our transcriptome. Column explanations are listed in the second tab in this spreadsheet.

• Supplementary file 2. Clustering analysis for gene expression during regeneration. Mapped reads were analyzed independently for weighted gene correlation network analysis (WGCNA). Column explanations are listed in the second tab in this spreadsheet, but the modules for each differentially expressed contig are in column C. A greater number of genes were identified as being significantly

upregulated in this analysis compared to our initial study because this list also includes genes differentially regulated in either direction (upregulation or downregulation) between any two samples. Also, the thresholds for fold-change and significance were not as strict.

• Supplementary file 3. (A) Expression patterns of upregulated genes cloned in this study. Expression patterns are as follows: BB- brain branch; Ep- epithelial; Ey- eyespots; G- gut; GC- germ cell; H-head; L- lateral margin; M- mesenchymal (parenchymal) expression pattern (could include muscle-like or neoblast-like patterns); N- neural; N+- neural plus another expression pattern (including neural with background); NS- neural subset; Ph- pharynx; Pol- polar expression; Pr- protonephridia; Pu-punctate; U- ubiquitous. (B) Genes with functions in regeneration. Includes genes with small brain regeneration phenotypes, plus *arrowhead* and *SP6* which functions in eye regeneration. The results of ISH experiments, phenotype (including whether significant result was found in brain regeneration quantification), and follow-up RNAi experiments with *sFRP-1* and *smedwi-1* are shown. For *sFRP-1* and *smedwi*-1 experiments, the results are summarized as follows: P- signal present; R- signal reduced; A- signal absent. For each contig, results from clustering analysis, expression in FACs analysis (*LabbÉLabbé et al., 2012*), and time points for significant upregulation vs. cut control ($\geq$2x, p$\leq$0.01) are also shown. BLAST hits and other annotation are also presented. (C) Genes with functions in homeostasis. Includes 26 genes that caused phenotypes during homeostasis in RNAi experiments. The table includes similar information given in *Supplementary file 3B*. (D) Genes with no phenotypes. Summary of all other genes from this analysis, including those that gave no detectable phenotype after RNAi. (E) List of genes used for qPCR experiments in this study. (F) Total preliminary and final qPCR results from *Figure 3*. Each tab includes results for one gene in our study. (G) Homologs of genes from this study. (H) Gene expression pattern for downregulated genes. Here we present a list of 71 downregulated genes from this study, including the expression patterns for these genes. (I) Annotation information and primers for genes described in this study.

### Major datasets

The following dataset was generated:

| Author(s) | Year | Dataset title | Dataset URL | Database, license, and accessibility information |
|---|---|---|---|---|
| Roberts-Galbraith, Brubacher, Newmark | 2016 | Illumina Sequencing of transcripts during regeneration | https://www.ncbi.nlm.nih.gov/sra/?term=PRJNA319973 | Publicly available at the Sequence Read Archive (accession no: PRJNA319973) |

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
