## [Decision Letter]

Thank you for submitting your article "A functional genomics screen in planarians reveals regulators of whole-brain regeneration" for consideration by *eLife*. Your article has been favorably evaluated by Fiona Watt (Senior Editor) and two reviewers, one of whom, Alejandro Sánchez Alvarado (Reviewer #1), is a member of our Board of Reviewing Editors.

The reviewers have discussed the reviews with one another and the Reviewing Editor has drafted this decision to help you prepare a revised submission.

Summary:

Roberts-Galbraith and colleagues report on an ambitious effort to identify genes involved in planarian cephalic regeneration, particularly those associated with the nervous system of these organisms. By analyzing changes in gene expression, selecting those candidate genes with the most significant changes and then following up on these candidates by both RNAi and whole-mount visualization of gene expression, the authors have produced an extensive map of genetic activities associated with brain function and regeneration in planarians. For instance, the authors report that *soxB-2a* controls production of neural progenitors, that *hesl-3* is important for expressing patterning genes marking the far anterior region, that *mblk* suppresses expression of several neural transcription factors. Given the nature and design of the screen, it is not surprising that the findings reported here would encompass a very broad range of biological functions that not only expands on prior work in the field but also contributes novel and interesting findings to our understanding of neuronal regeneration. Chiefly among these, are the discovery of planarian glial-like cells, the identification of genes regulating regeneration and of specific cephalic ganglia regions likely associated with chemosensation in this animal.

Overall, this is a comprehensive screen reporting on a large number of novel molecules labeling cells (glia), cephalic regions and domains, as well as a number of intriguing phenotypes. However, the analyses of all the reported findings are uneven in depth and complexity, which is to be expected by the very nature of the approach. Because this work opens many new avenues of research, rather than recommending an even expansion of the analyses for all of the genes reported, it is our opinion that both the authors and the developmental biology and planarian communities would be best served if these findings are reported as an *eLife* Tools and Resources article.

Essential revisions:

1) There is a marked discordance between what is presented in the Introduction and Discussion and what is being reported in the manuscript. In fact, this work would benefit greatly if the authors were to provide a significantly more expanded background elaborating in more detail what was known in the field prior to the screen results reported here, particularly in those areas in which the authors report discoveries. As it stands, neither the Introduction nor the Discussion provide the necessary detailed information to place the discoveries reported here in their proper context. I would like to suggest that the authors rewrite the Introduction and Discussion background sections accordingly so as to shine a brighter light on the significance of their findings.

2) Figure 2 and *hesl-3*Figure 3: It is not clear from the 18/28 genes required for brain size but not expression of *sFRP-1* or *smedwi-1* that these specifically control brain size (subsection “An unbiased functional screen reveals genes with roles in planarian brain regeneration”, third paragraph) as only effects on anterior regeneration are described. Do these animals have reduced anterior and posterior blastema size? One would anticipate that bona fide brain regulators only minimally disrupt regeneration of other structures.

3) The use of qPCR to phenotype animals in this screen is a valuable starting point for analysis that ideally would be confirmed by staining, as decreases in animal-wide transcript abundance could be reflective of uniform or localized reduction of expression. The main conclusions of this analysis should be verified with in situ hybridizations. In particular, the function of *soxB-2a* in producing neural progenitors and the function of *hesl-3* in promoting expression in the far anterior, and the function of *mblk* to suppress expression of castor and *lhx1/5-1*. If this is not possible, the authors should address this matter in the text of the manuscript.

4) A fraction of the 134 gene qPCR data (~8 transcripts for each condition) is displayed to describe expression changes in *soxB-2a, runt, hesl-3, mblk* and *CRELD* phenotypes but not others. The remaining data should be included, in particular to support the specificity of the expression phenotypes shown.

5) The *LDLRR* data is intriguing but not well developed as the other findings reported here. Several outstanding questions remain regarding this interesting class of molecules. Do these genes promote regeneration from any type of injury? As injury-induced genes, do they affect expression of follistatin or other early injury-induced genes? Do *LDLRR-1* and *-3* function redundantly? We would suggest either elaborating to determine the function of these genes in regeneration or omitting so that such analysis can be part of its own report.

6) We are particularly intrigued by the behavioral assay reported in this manuscript, as it is an important contribution to the field likely to open many new avenues for research. As such, it seems important for the authors to include data underscoring both the specificity of the RNAi phenotypes and the assay. Including data in which, for example, the *FLI-1* phenotypes do not display aberrant locomotion would help in this regard.

---

## [Author Response]

*Overall, this is a comprehensive screen reporting on a large number of novel molecules labeling cells (glia), cephalic regions and domains, as well as a number of intriguing phenotypes. However, the analyses of all the reported findings are uneven in depth and complexity, which is to be expected by the very nature of the approach. Because this work opens many new avenues of research, rather than recommending an even expansion of the analyses for all of the genes reported, it is our opinion that both the authors and the developmental biology and planarian communities would be best served if these findings are reported as an eLife Tools and Resources article.*

We have added experiments in several sections to flesh out our findings. We believe that the contributions of this manuscript tend more toward elucidation of biological phenomena and less toward the production of a dataset. As such, we hope that the reviewers will consider revisiting this issue based on new data in the manuscript and the edits made in writing (see below). However, should the reviewers still feel that this manuscript fits best as a Tools and Resources article, we will respect that decision.

*Essential revisions:*

*1) There is a marked discordance between what is presented in the Introduction and Discussion and what is being reported in the manuscript. In fact, this work would benefit greatly if the authors were to provide a significantly more expanded background elaborating in more detail what was known in the field prior to the screen results reported here, particularly in those areas in which the authors report discoveries. As it stands, neither the Introduction nor the Discussion provide the necessary detailed information to place the discoveries reported here in their proper context. I would like to suggest that the authors rewrite the Introduction and Discussion background sections accordingly so as to shine a brighter light on the significance of their findings.*

We have edited both the Introduction and Discussion to better contextualize the forward progress we have made toward addressing several key questions. We appreciate this especially helpful feedback and hope that the revised text makes our contributions and their significance clearer for all readers. We have attached a copy of our manuscript with tracked changes; the extensive revisions are clearly visible in this draft.

*2) Figure 2 and hesl-3 Figure 3: It is not clear from the 18/28 genes required for brain size but not expression of sFRP-1 or smedwi-1 that these specifically control brain size (subsection “An unbiased functional screen reveals genes with roles in planarian brain regeneration”, third paragraph) as only effects on anterior regeneration are described. Do these animals have reduced anterior and posterior blastema size? One would anticipate that bona fide brain regulators only minimally disrupt regeneration of other structures.*

We repeated RNAi of the 27/30 genes^*^ which cause small brain regeneration phenotypes (excluding the 3/30 RNAi conditions we already showed cause neoblast defects) and quantified the size of both anterior and posterior blastemas. The results of these experiments are shown in columns E-F of [Supplementary-material SD3-data]. We saw significantly smaller blastemas after RNAi targeting *tRNA synthetase, GTPbp, argininosuccinate synthase, wdr24*, and *smarcb1* genes. These results are also included in Figure 2 (grey diagonal lines on the bars). For all but *GTPbp*, small blastemas were seen in both anterior and posterior directions. One gene (*hesl-3*) caused a slight, but significant, increase in head blastema size. Taken together, we conclude that 19/30 genes with small brain regeneration phenotypes are not required for neoblast maintenance, polarity reestablishment, or blastema formation. This suggests some level of specificity for our screen, and in particular for most of the genes on which we focused in our later analyses.

^*^The number of genes in this category changed slightly between submissions based on repetition of some brain regeneration quantification experiments for which we initially had too few animals for results to be supported by adequate statistical power.

*3) The use of qPCR to phenotype animals in this screen is a valuable starting point for analysis that ideally would be confirmed by staining, as decreases in animal-wide transcript abundance could be reflective of uniform or localized reduction of expression. The main conclusions of this analysis should be verified with in situ hybridizations. In particular, the function of soxB-2a in producing neural progenitors and the function of hesl-3 in promoting expression in the far anterior, and the function of mblk to suppress expression of castor and lhx1/5-1. If this is not possible, the authors should address this matter in the text of the manuscript.*

qPCR is quantitative, whereas in situ hybridization (as we are currently doing these experiments) is not. Thus a 25% reduction in a given transcript might not be easily seen by ISH, especially given that signal amplification is inherent in the ISH process and several factors influence signal intensity besides transcript number.

In spite of this caveat, we have completed selected follow-up experiments for *soxB2-2* and *hesl-3* as requested by the reviewers. Although its signal is very difficult to detect (due to its low abundance: RPKM = 0.2 in whole animals), we did see *hlh-1* cells in *control(RNAi)* but not *soxB2-2^**^(RNAi)* animals (Figure 3—figure supplement 1). We attempted to examine *otxA* and *POU2/3-2* as well; these markers were not absent in the brain after *soxB2-2(RNAi)*, but both of these genes are expressed more broadly than *hlh-1*. Likely, *soxB2-2* is only affecting *otxA* and *POU2/3-2* expression in a subset of cells (either in the progenitor pool or in the brain), but interpretation is difficult without more markers. We have thus omitted the results of these experiments.

The results of *sFRP-1* and *foxD* ISH after *hesl-3(RNAi)* are shown in Figure 2—figure supplement 1 and Figure 3—figure supplement 1. These markers were not absent after *hesl-3(RNAi)*, but were decreased consistent with the 40-50% decrease seen by qPCR.

*^**^*We have changed the name of *soxB-2a* to *soxB2-2* to be more consistent with nomenclature conventions and to align our gene with that described in *S. polychroa* (Monjo & Romano, 2014).

*4) A fraction of the 134 gene qPCR data (~8 transcripts for each condition) is displayed to describe expression changes in soxB-2a, runt, hesl-3, mblk and CRELD phenotypes but not others. The remaining data should be included, in particular to support the specificity of the expression phenotypes shown.*

Full raw data are now included in [Supplementary-material SD3-data]. The expression of most genes was not altered by each perturbation.

*5) The LDLRR data is intriguing but not well developed as the other findings reported here. Several outstanding questions remain regarding this interesting class of molecules. Do these genes promote regeneration from any type of injury? As injury-induced genes, do they affect expression of follistatin or other early injury-induced genes? Do LDLRR-1 and -3 function redundantly? We would suggest either elaborating to determine the function of these genes in regeneration or omitting so that such analysis can be part of its own report.*

We have included additional data on *LDLRR-1/3* and *F-spondin*. For each of these genes, RNAi did not cause small blastemas (see comment above and [Supplementary-material SD3-data]), suggesting that their role in regeneration is somewhat specific. We examined the effect of *LDLRR-1(RNAi), LDLRR-3(RNAi)*, and *F-spondin(RNAi)* on the expression of *follistatin, jun, inhibin*, and *runt* (injury-induced genes as per Wenemoser, et al. 2012) 6 hours after both anterior and posterior amputation (Figure 4—figure supplement 1). All gene expression appeared normal, indicating that wound response is also not generally affected by these perturbations. We knocked down *LDLRR-1* and *-3* together and did not see an enhancement of the brain regeneration phenotype; however, we have discovered that *LDLRR-1* and *-3* are in a family with several other similar genes. It is possible that there is synergism or redundancy between multiple members of this family, but it will take significantly more work to explore this possibility, so we have not included our preliminary data on this subject. We also confirmed that these genes are not downregulated after irradiation, with *F-spondin*and *LDLRR-3*cells still present at 7 d post-irradiation when the worms showed significant lysing and tissue damage (Figure 4—figure supplement 1). This result supports our conclusion that *F-spondin* and *LDLRR-3* are present in cells that are not neoblasts or their direct descendants. Finally, we completed a pilot qPCR experiment after *LDLRR-3(RNAi)* to determine whether we could identify pathways or cell types affected by this gene perturbation. However, we did not see any genes for which expression was dramatically altered by *LDLRR-3(RNAi)*, indicating that a deeper investigation would be required to form a hypothesis about what these molecules are doing.

In all, we do believe that our results indicate the existence of parenchymal cells and cues that promote CNS regeneration. Because we know so little about the cues that promote and permit regeneration in planarians and otherwise, we believe that these findings are of value and should be included in this publication. In rewriting the Introduction and Discussion, we have attempted to outline the significance of this finding to more clearly communicate its importance.

*6) We are particularly intrigued by the behavioral assay reported in this manuscript, as it is an important contribution to the field likely to open many new avenues for research. As such, it seems important for the authors to include data underscoring both the specificity of the RNAi phenotypes and the assay. Including data in which, for example, the FLI-1 phenotypes do not display aberrant locomotion would help in this regard.*

We added a movement assay supporting normal locomotion in *FLI-1(RNAi)* animals (Figure 6—figure supplement 1).